# Spectral attenuation of ocean waves in pack ice and its application in calibrating viscoelastic wave-in-ice models

Sukun Cheng[1,4], Justin Stopa[2], Fabrice Ardhuin[3] and Hayley H. Shen[4]

[1]Nansen Environmental and Remote Sensing Center, Bergen, Norway

[2]Department of Ocean and Resources Engineering, University of Hawaii, Mānoa, HI, USA

[3]Univ. Brest, CNRS, IRD, Ifremer, Laboratoire d'Océanographie Physique et Spatiale (LOPS), IUEM, Brest, France

[4]Department of Civil and Environmental Engineering, Clarkson University, Potsdam, NY, USA

*Correspondence to*: Sukun Cheng (sukun.cheng@nersc.no)

Three key points:

1. The spatial distribution of wavenumber and spectral attenuation in pack ice are analyzed from SAR retrieved surface wave spectra.

2. Spectral attenuation rate of 9~15s waves varies around $10^{-5}m^2/s$, with lower values in thicker semi-continuous ice field with leads.

3. The calibrated viscoelastic parameters are greater than those found in pancake ice.

**Abstract.** We investigate a case of ocean waves through a pack ice cover captured by Sentinel-1A synthetic aperture radar (SAR) on 12 October 2015 in the Beaufort Sea. The study domain is 400 km by 300 km adjacent to a marginal ice zone (MIZ). The wave spectra in this domain were reported in a previous study (Stopa et al. 2018b). In which, the authors divided the domain into two regions delineated by the first

appearance of leads (FAL) and reported a clear change of wave attenuation of the total energy between the two regions. In the present study, we use the same dataset to study the spectral attenuation in the domain. According to the quality of SAR retrieved wave spectrum, we focus on a range of wavenumbers corresponding to 9~15 s waves from the open water dispersion relation. We first determine the apparent attenuation rates of each wavenumber by pairing the wave spectra from different locations. These attenuation

rates slightly increase with increasing wavenumber before the FAL and become lower and more uniform against wavenumber in thicker ice after the FAL. The spectral attenuation due to the ice effect is then extracted from the measured apparent attenuation and used to calibrate two viscoelastic wave-in-ice models. For the Wang and Shen (2010) model, the calibrated equivalent shear modulus and viscosity of the pack ice are roughly one order of magnitude greater than that in grease/pancake ice reported in Cheng et al. (2017).

These parameters obtained for the extended Fox and Squire model are much greater, as found in Mosig et al. (2015) using data from the Antarctic MIZ. This study shows a promising way of using remote sensing data with large spatial coverage to conduct model calibration for various types of ice cover.

KEY WORDS: pack ice, ocean waves, SAR derived wave spectra, wave attenuation, viscoelastic model calibration

## 1 Introduction

Rapid reduction of Arctic ice in recent decades has become a focal point in climate change discussions (Comiso et al., 2008; Meier and Thomas, 2017; Rosenblum and Eisenman, 2017; Stroeve and Notz, 2018). The reduction emphasizes the need to better understand the complex interaction between the sea ice, the ocean, and the atmosphere. One of these interaction processes is between ocean waves and sea ice. Ocean waves help to shape the formation of new ice covers (Lange et al., 1989; Shen et al., 2001), break existing ice covers (Kohout et al., 2016), modify the upper ocean mixing (Smith et al., 2018), or potentially compress sea ice through wave radiation stress (Stopa et al., 2018a). In turn, ice covers suppress wave-wind interaction by reducing the fetch. They also alter the wave dispersion and attenuation through scattering and dissipation (Squire, 2007, 2018, 2020).

Modeling surface gravity waves in polar oceans requires the knowledge of many source terms. These source terms include wind inputs and dissipation, nonlinear transfer between frequencies, and wave-ice interaction. WAVEWATCH III® (WW3, WAVEWATCH III® Development Group, 2019), one of the most widely used third-generation wind-wave models for global and regional wave forecasts, has implemented several dispersion/dissipation (IC0, IC1, IC2, IC3, IC4, and IC5) and scattering (IS1 and IS2) parameterizations, called "switches" in WW3, to estimate the ice effect on waves. Of the two scattering switches, IS1 redistributes a constant fraction of the incoming wave energy to all directions isotropically. IS2 adopts a linear Boltzmann equation and an estimation of maximum floe diameter due to ice breakup to model wave scattering, combined with a creep related dissipation. As will be discussed in section 3.2, scattering is negligible in the present dataset, hence will not be considered in the present study. Of the six different dispersion/dissipation parameterizations, IC0 and IC1 are out-of-date. IC4 is empirically determined from data obtained in the Southern Ocean. IC2 is based on a theory where wave damping is entirely due to the eddy-viscosity in the water body beneath the ice cover. The other two parameterization, IC3 and IC5 both assume that wave damping is due to the processes within the ice cover alone. In this study, we will focus on IC3 and IC5 parameterizations. The same method may be applied to calibrate IC2. Both IC3 and IC5 theorize that sea ice can store and dissipate mechanical energy, hence they model the ice cover as a viscoelastic material. The storage property is reflected in the potential and elastic energy, and the dissipative property is in the equivalent viscous damping. The difference between IC3 and IC5 is that IC3 is an extension of the viscous ice layer model with a finite thickness (Keller, 1998) by including elasticity into a complex viscosity (Wang and Shen, 2010), while IC5 is an extension of the thin elastic plate model (Fox and Squire, 1994) introduced by Mosig et al. (2015) by adding viscosity into a complex shear modulus (equivalent to the complex viscosity via the Voigt model). Below we will refer to these two viscoelastic models as WS and FS respectively. Each of the two models shows a frequency-dependent wave propagation determined by the dispersion relation. This relation, which depends on the viscoelastic parameters, specifies how the wavenumber $k = k_r + ik_i$ is related to the wave frequency $f$. The complex wavenumber $k$ contains a real part $k_r$ which determines the wave speed and an imaginary part $k_i$ which determines the damping rate. To use these models for wave forecasts in ice-covered seas, one needs to determine the viscoelastic parameters

for all types of ice covers. To derive these parameters using first principles is challenging, as demonstrated by de Carolis et al. (2005), who obtained the viscosity of grease ice using principles in fluid mechanics of a suspension. Alternatively, an inverse method has been adopted to parameterize IC3. Using in-situ data from

the R/V Sikuliaq field experiment (Thomson et al., 2018), the WS model was calibrated to match the observed wavenumber and attenuation (Cheng et al., 2017). This calibration was carried out in a marginal ice zone (MIZ) populated predominantly with grease/pancake ice. Although it showed good agreement between the calibrated model and the field data in the frequency band containing most of the wave energy, these calibrated values are limited to the grease/pancake ice type. Further into the ice cover, where more rigid ice with larger

floes is present, how the viscoelastic parameters might change is unknown at present. Note that models can only be as robust as the training data. Therefore, using a broader type of sea ice data will result in a more robust model.

Advancements made in remote sensing technology have provided opportunities for such model calibration. Ardhuin et al. (2017) developed a method to conditionally invert from ice-covered water wave orbital motion

to directional wave spectra from the synthetic aperture radar (SAR) images based on the velocity bunching mechanism. The methodology is furthered improved in Ardhuin et al. (2018) and Stopa et al. (2018b) to study wave state in ice-covered Beaufort Sea using SAR images, which were captured in the R/V Sikuliaq field experiment by the satellite Sentinel-1A. Stopa et al. (2018b) retrieved wavenumber-dependent spectra under the partially and fully ice-covered regions by substantially reducing data contamination by ice features with

similar length scale as the wavelength. Using the SAR retrieved wave spectra, Stopa et al. (2018b) found that the significant wave height attenuated steeply prior to the first appearance of ice leads (denoted as FAL hereafter), with milder attenuation rate after the FAL. The definition of FAL is described in Appendix A. Furthermore, Monteban et al. (2019) used two overlapping burst images from the Sentinel-1 separated by ~ 2 s to investigated wave dispersion in Barents Sea. Based on the method proposed by Johnsen and Collard

(2009), this time separation between subsequent images was sufficient to result in a less noisy and higher quality imaginary spectrum, therefore, allowed them to obtain spatiotemporal information of the dispersion relation. Their results showed that for long waves (100m~350m) in thin ice (<40 cm) the dispersion relation was the same as in open water.

This study uses the dataset reported in Stopa et al. (2018b). It includes two parts of data analysis: obtaining

spectral wave attenuation rates from the retrieved wave data, and then use these attenuation rates for wave-in-ice model calibration. This paper is organized as follows: section 2 introduces wave spectra data from Stopa et al. (2018b) used in this study. From this data, we obtain the dominant wavenumbers and the related wave directions over the studied domain. In section 3, we use the directional wave spectra to derive the apparent attenuation rate and then the attenuation rate due to sea ice alone. In section 4, we calibrate two

viscoelastic wave-in-ice models using the obtained wavenumber-dependent attenuation data. The methodology presented in these two sections is similar to that in Cheng et a. (2017) with modifications to resolve the difference of wave spectral data types. In Cheng et al. (2017), the spectral data was between energy and frequency, while in Stopa et al. (2018b) it was between energy and wavenumber. Section 5

discusses the characteristics of wavelength and attenuation in pack ice, calibrated viscoelastic parameters, and wave attenuation modeling. The final conclusions are given in section 6.

## 2 Data description

During the R/V Sikuliaq experiment, the Sentinel-1A equipped with a synthetic aperture radar (SAR) acquired six sequential images around 16:50 UTC on 12 October 2015. These SAR images covered a 400 km by 1,100 km region including open water, grease/pancake ice, and pack ice (refer to Figure 1 in Stopa et al. (2018b)). A large wave event during this time with wave heights exceeding 4 m in the captured region provided quality wave data. From these SAR images, Stopa et al. (2018b) obtained two-dimensional wave spectra data $E(k_x, k_y)$ for most part of this ice-covered region, where $E$ indicates wave energy density, $k_x$ and $k_y$ are the wavenumber components in the range and azimuth directions of the satellite track, respectively. Details of this dataset and its retrieval may be found in Stopa et al. (2018b). We convert the two-dimensional spectrum at each location into an equivalent wavenumber-direction spectrum $E(k_r, \theta)$, where $k_r = \sqrt{k_x^2 + k_y^2}$ and $\theta = \text{atan}(k_y/k_x)$ indicate wavenumber and direction, respectively. The wavenumber is discretized from 0.011 m$^{-1}$ to 0.045 m$^{-1}$ with an increment of 0.002 m$^{-1}$, and the direction is discretized into 360 bins with a 1$^\circ$ bin width. We then define wavenumber-dependent main wave directions for each given spectrum, $E(k_r, \theta)$ in the following way. For each wavenumber $k_r$ we fit the corresponding $E$ curve with a Gaussian function, where an example is given in Figure S1 of the supplemental material. The mean of this Gaussian function is defined as the main wave direction $\theta_{k_r}$ for this $k_r$. Furthermore, we define the dominant wavenumber, $k_{r,dominant}$, to be the one corresponding to the maximum directionally integrated wave energy $\int E(k_r, \theta) \, d\theta$.

An overview of the processed wave conditions in terms of the dominant wavenumber and its main direction is given in Figure 1(a), along with the locations of the ice edge and other in-situ observations. The associated ice conditions in the region are presented in Figures 1(b)(c). Significant spatial variability is observed in both the wave and ice conditions. Figure 1(a) presents a subregion captured by the SAR images covering from a portion of Alaska to the last azimuth position where waves are detectable in the images. Colors indicate the dominant wavenumber distribution of the retrievals, and arrows indicate their main directions. Cell size is coarsened to 12.5 km $\times$ 12.5 km to enhance visualization. The ice edge is indicated by the contours of ice concentration (<0.4) from AMSR2 (Advanced Microwave Scanning Radiometer 2, http://doi.org/10.5067/AMSR2/A2_SI12_NRT). An in-situ buoy: AWAC-I (a subsurface Nortek Acoustic Wave and Current buoy, moored at 150$^\circ$W, 75$^\circ$N) is marked by a magenta asterisk. Except for the in-situ observations from the Sikuliaq ship (green diamond) and several drifting buoys (blue dots) near the ice edge, ice morphology information including ice types and their partial concentrations are absent. Nevertheless, the FAL (red dots) presumably marks the separation between discrete floes and a semi-continuous ice cover with dispersed leads. The ice condition below (before) the FAL was more complex with thinner ice and lower concentration than that above (after) the FAL.

We select the study domain defined by the azimuth from 450 to 750 km and the range from 0 to 400 km. This domain contains most of the wave data retrieved in the pack ice field. Figures 1(b)(c) show the distributions of ice concentration from AMSR2 and ice thickness from SMOS (Soil Moisture and Ocean Salinity, https://icdc.cen.uni-hamburg.de/1/daten/cryo-sphere/l3c-smos-sit.html), respectively, in the region of interest. As shown in Cheng et al. (2017) (Supporting information Figures S6 and S7), these two ice products compared the best with in-situ observations in the MIZ. Their accuracies in the pack ice zone are uncertain. The purpose of this work is to retrieve the spectral wave attenuation and use the result to calibrate viscoelastic models in regions dominated by thin pack ice (thickness<0.3 m). As shown in Figure 1(a), $k_{r,dominant}$ generally declines crossing the FAL towards the north. Before the FAL, $k_{r,dominant}$ increases in the direction of increasing range and decreasing ice concentration but is insensitive to ice thickness variation. After the FAL, $k_{r,dominant}$ decreases in the wave propagating direction (arrows) associated with the increase of ice thickness, where the ice field is presumably a semi-continuous cover populated with leads.

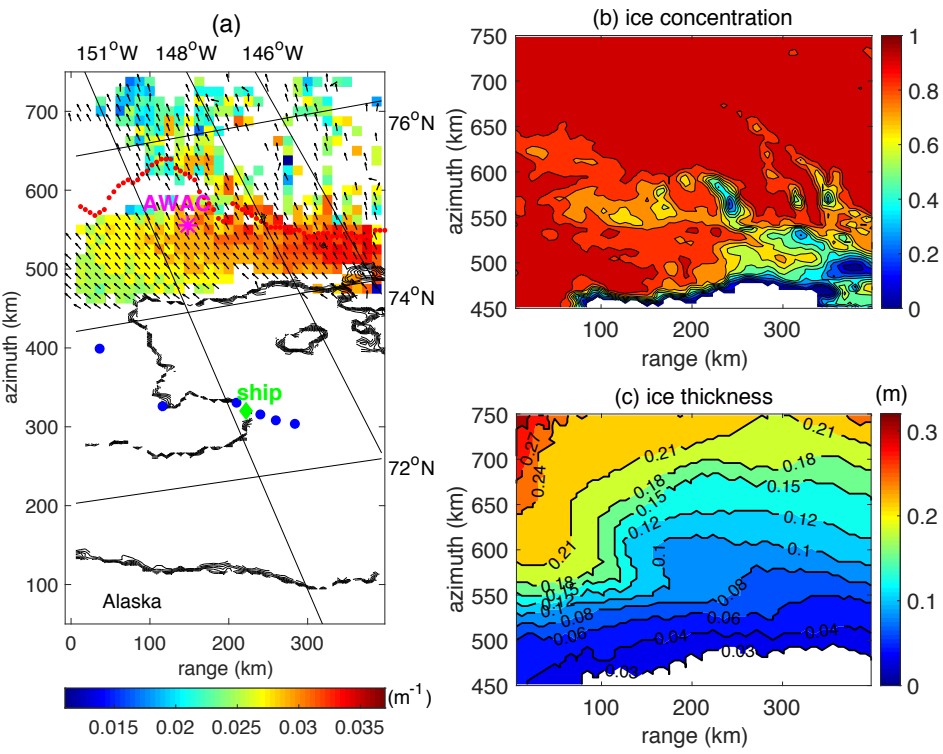

**Figure 1. (a) Overview of the retrieved data distribution around 16:50 UTC on 12 October 2015. Colors represent the dominant wavenumber and arrows represent the main direction of the dominant wavenumber. Red dots indicate the first appearance of leads (FAL). Locations of the Sikuliaq ship and the buoys operating at that time are indicated by green diamond and blue dots, respectively. AWAC-I is marked by a magenta asterisk. Ice edges are indicated by contours of ice concentration < 0.4 from AMSR2 (b)(c) Distributions of ice concentration (AMSR2) and ice thickness (SMOS) in the selected region, respectively.**

$d\theta$ Figures 2(a)(b) show two-dimensional histograms of the main wave direction for each wavenumber $\theta_{k_r}$ before and after the FAL, respectively, where wave direction is defined in the meteorological convention (i.e. the direction 'from' in degrees clockwise from True North). Grayscale indicates the occurrence frequency of

$\theta_{k_r}$ in each $k_r$ bin. We observe a significant change of $\theta_{k_r}$ crossing the FAL: $\theta_{k_r}$ before the FAL spreads from 160º to 190º, while $\theta_{k_r}$ is more tightly clustered from 180º to 200º after the FAL. The difference before and after the FAL is most significant for $k_r < 0.035$ m$^{-1}$

For the subsequent spectral analysis, we further restrict $k_r$ to 0.019 m$^{-1} \leq k_r \leq \min(2\pi/\lambda_c, 0.045\text{m}^{-1})$, where $\lambda_c$ is the azimuth cutoff indicating the minimum resolvable wavelength from the SAR imagery (Stopa et al., 2015; Ardhuin et al., 2017). Below this wavelength, the patterns of ice-covered ocean surface roughness from SAR imagery are more related to ice features rather than waves (Stopa et al., 2018b). For $k_r < 0.019$ m$^{-1}$, energy density $E(k_r, \theta_{k_r})$ is small with high spatial variation (Figure B1 in Appendix B), hence treated as

noise band and removed from further study.

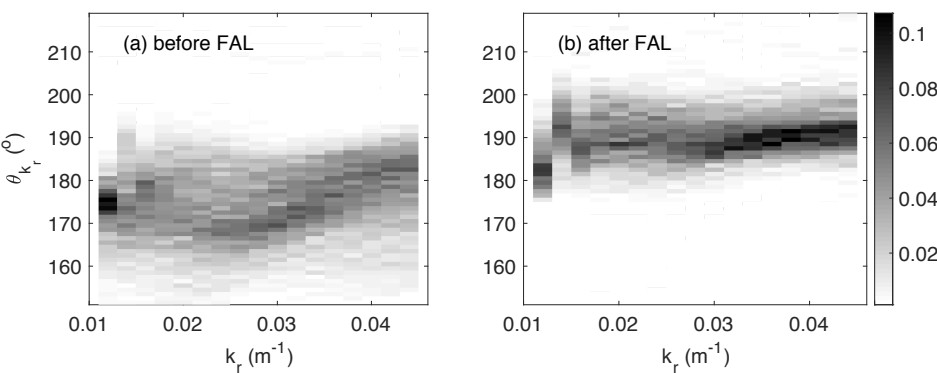

**Figure 2. Two-dimensional histogram of $\theta_{k_r}$ and $k_r$ collected (a) before and (b) after the FAL. Grayscale represents the occurrence frequency of $\theta_{k_r}$ in each $k_r$ bin.**

The corresponding frequency $f$ (wave period $T$) range is estimated as 0.067 to 0.11 Hz (9 to 15 s) using the

open water dispersion relation, $(2\pi f)^2 = g k_{ow}$, where $g$ is the gravitational acceleration, $k_{ow}$ indicates the wavenumber in open water. The actual dispersion relation which is likely dependent on the ice condition cannot be measured from our instantaneous SAR data. Using accelerometers deployed on ice floes (Fox and Haskell, 2001) and from marine radar or buoy data (Collins et al., 2018), it was found that the wavenumber in ice-covered region within about 10 km from the ice edge was close to that in open water. Further into the

ice cover, Monteban et al. (2019) used two overlapping burst images from the Sentinel-1 separated by ~ 2 s to investigated wave dispersion in the Barents Sea. The ice concentration and thickness in their study were similar to the present study. The computed wave dispersion within the sea ice for long waves (peak wavelengths > 100 m) was in good agreement with the open-water dispersion relation.

### 3 Wave attenuation

In this section, we first obtain the apparent spectral wave attenuation by pairing the directional spectra at different locations. We then remove the contributions of wave energy between pairs of locations from wind input, wave breaking dissipation, and nonlinear transfer between frequency components to extract the spectral

attenuation due to ice effects alone. These spectral attenuation rates due to ice will be used in section 4 to calibrate two viscoelastic wave-in-ice models.

## 3.1 Apparent wave attenuation

We define an apparent wave attenuation for each $k_r$ by assuming exponential decay of the wave spectral densities from location $A$ to location $B$:

$$\alpha(k_r) = \frac{1}{2Dcos(|\bar{\theta}-\theta_{AB}|)} ln\left(\frac{E_A(k_r,\bar{\theta})}{E_B(k_r,\bar{\theta})}\right) \tag{1}$$

where $\bar{\theta} = \frac{\theta_{kr,A}+\theta_{kr,B}}{2}$ is the average of the main wave directions at $A$ and $B$; $D$ and $\theta_{AB}$ are the distance and direction from $A$ to $B$ in the longitude-latitude coordinates, respectively. A selected pair of $A$ and $B$ is named as a pair hereafter. To reduce the uncertainties of naturally present ice and wave variability, a set of quality control criteria are applied to a pair before further analysis:

1) Ignore substantially oblique waves. The difference between $\bar{\theta}$ and the vector from location $A$ to location $B$ is restricted to $|\bar{\theta} - \theta_{AB}| \leq 15°$.
2) Avoid strong spatial variations of ice condition between $A$ and $B$. Distance between $A$ and $B$ is restricted to $D \leq 60$ km.
3) As the wave state changes significantly across the FAL as mentioned in section 2, no pair across the FAL is selected. This criterion enables us to detect, if any, the influence of ice morphology on wave attenuation.
4) Ensure point $A$ and $B$ are both subject to the same wave system, the Pearson correlation coefficient between energy spectra $E_A(k_r,\theta)$ and $E_B(k_r,\theta)$ is required to be greater than 0.9. The Pearson correlation coefficient is defined as $r = \frac{\Sigma_i(x_i-\bar{x})(y_i-\bar{y})}{\sqrt{\Sigma_i(x_i-\bar{x})^2}\sqrt{\Sigma_i(y_i-\bar{y})^2}}$, where $x_i$ and $y_i$ are the PSD values at the $i^{th}$ wavenumber.
5) Exclude outliers of the spectral attenuation where wave energy of B is higher or close to that of $A$. Thus, $\alpha(k_r) > 10^{-6}$ m$^{-1}$ is required.
6) A selected pair has at least 10 data points of $\alpha(k_r)$ to do calibration in section 4.

We obtain 2634 pairs (2194 pairs before the FAL, and 440 pairs after the FAL) through the above criteria to calculate the apparent wave attenuation $\alpha(k_r)$ by Eq. (1). The results are sorted statistically to show the occurrence frequency of $\alpha(k_r)$. The $\alpha$ domain is equally divided into 30 bins from $10^{-6}$ to $10^{-4}$ m$^{-1}$ in log scale, and the $k_r$ domain is equally divided from 0.011 m$^{-1}$ to 0.0045 m$^{-1}$ with an increment of 0.002 m$^{-1}$ as mentioned earlier. Figures 3(a)(b) show two-dimensional histograms of $\alpha$ against $k_r$ before and after the FAL, respectively. Grayscales represent the occurrence of $\alpha$ at each combined bin of $\alpha$ and $k_r$. Red curves indicate the most frequent occurrence of $\alpha(k_r)$ against $k_r$. It is observed that more data are obtained before the FAL, with a slightly increasing trend of $\alpha(k_r)$ versus $k_r$ before the FAL, while $\alpha(k_r)$ obtained after the FAL are mostly lower and independent of $k_r$.

The range of this apparent spectral attenuation is in agreement with Stopa et al. (2018b), in which, the authors selected multiple tracks throughout the ice region and focused on the overall decay of the significant wave height over hundreds of kilometers. The reduction of attenuation crossing the FAL shown in Figure 3 is also consistent with Stopa et al. (2018b), who reported a drop of attenuation of the significant wave height after the FAL. In Appendix B, we show the results of this long-range attenuation against wavenumber. (Figure

B1). The difference of attenuation obtained by the two methods, one based on short distances (<60 km) to reduce the effect of ice type variability and the other over long distances (~300 km) is discussed in section 5.

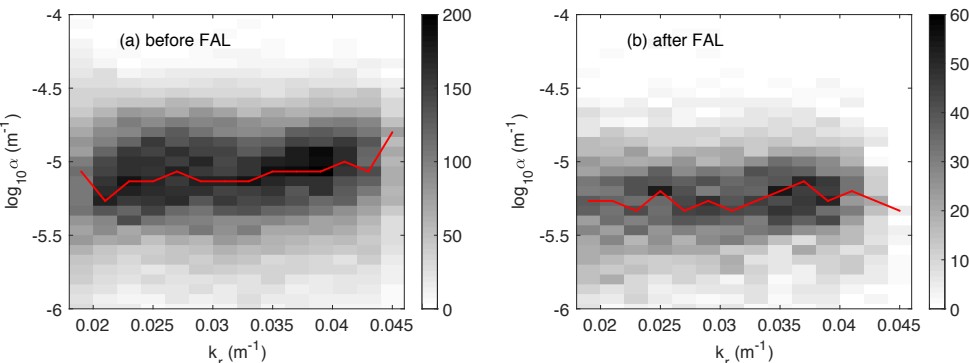

Figure 3. Smoothed two-dimensional histograms of wavenumber $k_r$ and the apparent attenuation $\alpha$, (a) before the FAL and (b) after the FAL. The $\alpha$ domain is equally divided into 30 bins in the log scale from $10^{-6}$ to $10^{-4}$ m$^{-1}$. The $k_r$ domain is divided from 0.011 m$^{-1}$ to 0.0045 m$^{-1}$ with an increment of 0.002 m$^{-1}$ Grayscale indicates the occurrence of $\alpha$ in each $\alpha$-$k_r$ bin from the selected pairs. The red curve indicates the highest occurrence of $\alpha$ against $k_r$.

Because of the large study domain and the apparent difference of $k_{r,dominant}$ in the east-west direction before FAL as shown in Figure 1(a), it is worthwhile examining the regional variability. Figure 4 displays the results related to $\alpha$ and $k_r$ in three subdomains from west to east bounded by longitudes: (150ºW, 151ºW), (148ºW, 149ºW) and (146ºW, 147ºW). The left column shows 467, 233 and 132 pairs selected in the three longitude bins, respectively. Each segment indicates a pair selected in section 3.1, with ends corresponding to the locations $A$, $B$ and its color indicates the mean ice thickness between the two. In the middle column, the $\alpha$ values corresponding to $k_r = 0.019, 0.025, 0.031$ and 0.039 m$^{-1}$ obtained from these pairs are separated into two groups: before and after the FAL. In each subdomain delineated by the longitude and the FAL, the distribution of $\alpha$ for each $k_r$ is presented by a violin plot, whose width indicates the probability density distribution of $\alpha$ and a circle marker inside indicates the median. These violin plots show that $\alpha$ before the FAL is larger than that after the FAL for all $k_r$ in all three subdomains. We note that the sample size after the FAL in the eastern-most subdomain (146ºW, 147ºW) is the lowest, corresponding to the largest variability of the violin plots. To track wave spectrum evolution from nearer the ice edge towards the interior ice zone, we collect wave power spectral densities (PSDs, $\int E(k_r,\theta)d\theta$) at the north end of each selected pairs in the left column. The averaged PSDs of this collection per 0.1 degree in latitudes are presented by curves in the right column, where colors indicate the latitude. As defined earlier, $k_r$ associated with the peak of a PSD curve is $k_{r,dominant}$. Before the FAL the magnitude of PSD drops rapidly while $k_{r,dominant}$ varies slightly as the latitude increases. In contrast, the PSDs change slowly after the FAL while $k_{r,dominant}$ decreases quickly. Note that at high latitudes, PSD at $k_{r,dominant}$(red) is higher than that of the same wavenumber at low latitudes. We will revisit this phenomenon in the discussion section.

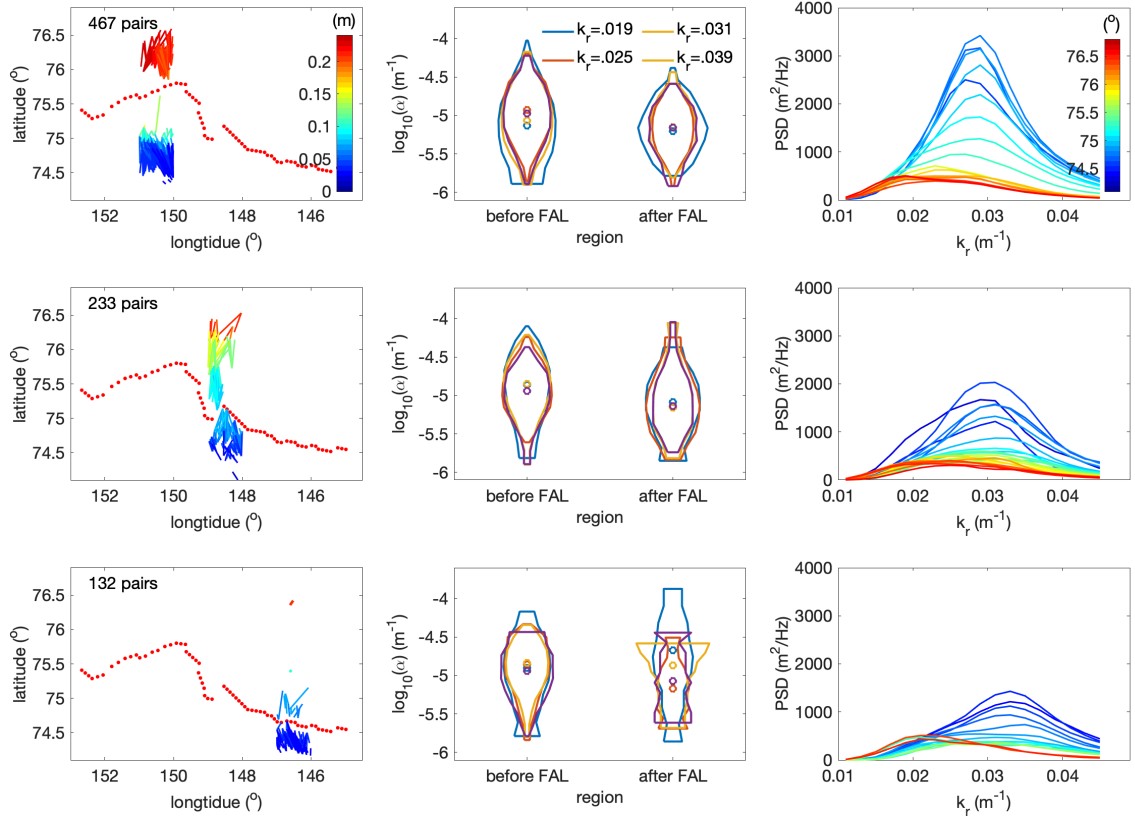

**Figure 4. Close view in three selected longitude intervals: (150ºW, 151ºW), (148ºW, 149ºW) and (146ºW, 147ºW). (Left column) Geographical distribution of the selected pairs, each of which is represented by a segment with color indicating the mean ice thickness. Red dots represent the FAL. (Middle column) Violin plots of the relevant $\alpha$ grouped by wavenumbers and before/after the FAL. (Right column) Evolution of the PSD of wave spectra per 0.1º in latitude. Note that in the rightmost interval (bottom row) the red PSD curves correspond to the few very short red segments near 76.5ºN.**

### 3.2 Wave attenuation due to ice effect

Following Eq. (1), the apparent attenuation obtained above is determined by the energy difference between two locations. This apparent attenuation is the result of multiple source terms, including the wind input $S_{in}$, damping through wave breaking and swell dissipation $S_{ds}$, the energy transfer due to nonlinear interactions among spectral components $S_{nl}$, and the dissipation/scattering of wave energy due to ice cover $S_{ice}$. In this section, we derive the attenuation rate due to $S_{ice}$ from the measured apparent attenuation $\alpha$.

The radiative transfer equation for surface waves concerning all the above effects is

$$\frac{\partial E}{\partial t} + \frac{\partial (c_g + U)E}{\partial x} = (1 - C)(S_{in} + S_{ds}) + S_{nl} + CS_{ice} \tag{2}$$

where $E = E(k_r, \theta, x, t)$ is the power spectral density depending on wavenumber, direction and location $x$; $c_g$ is the group velocity; $U$ is the current velocity. Note that $U$ in this region is below 0.1 m/s, according to OSCAR (Ocean Surface Current Analyses Real-time, https://www.esr.org/research/oscar/). Because the current speed is at least one order of magnitude below the estimated $c_g$ using the open water dispersion relation, we may drop it from Eq. (2). Furthermore, consistent with the fact that the dispersion relation in this

study is close to that of the open water, $c_g$ is relatively constant. Adopting the exponential wave decay along

$x$, i.e., $E(k_r, \theta, x) = E(k_r, \theta, x = 0)e^{-2\alpha x}$, we have $\frac{\partial(c_g+U)E}{\partial x} \approx c_g \frac{\partial E}{\partial x} = -2c_g \alpha E$.

Next, we examine the temporal derivative of wave energy $\frac{\partial E}{\partial t}$. It is challenging to calculate $\frac{\partial E}{\partial t}$ from the nearly

instantaneous SAR imagery. Instead, we estimate $\frac{\partial E}{\partial t}$ using hourly wave spectra data from two sources around

the time stamp of the SAR imagery: the in-situ AWAC-I marked in Figure 1 and the WW3 simulations of

the whole domain (REF run of Ardhuin et al., 2018). From the AWAC-I data, we obtain $\frac{\partial E}{\partial t}$ and compare that

with $c_g \frac{\partial E}{\partial x}$ using SAR retrieved wave data at the AWAC-I site. From the WW3 simulations, we obtain both

terms over the whole study domain. Both results consistently show that $\frac{\partial E}{\partial t}$ is at least two orders of magnitude

below $c_g \frac{\partial E}{\partial x}$. Thus, $\frac{\partial E}{\partial t}$ is dropped from Eq. (2).

     The other source terms $S_{in}$ and $S_{ds}$ are estimated using formulations from Snyder et al. (1981) and Komen et

al. (1984). For $S_{nl}$, we select the Discrete Interaction Approximation (DIA, Hasselmann et al., 1985a, b) to

estimate its value. Note that those parameterizations were formulated from open water study. How they might

change in the presence of ice covers is an open question. The formulations and associated coefficients used

here are described in Cheng et al. (2017). Wherever needed in these formulations, $f$ is approximated by the

open water dispersion relation with the measured $k_r$. Ice concentration is from AMSR2, ice thickness is from

SMOS, and wind data is from the Climate Forecast System Reanalysis (CFSR, Saha et al., 2010).

     Ice-induced wave attenuation is known from the dissipation of wave energy and scattering of waves (e.g.,

Wadhams et al., 1988; Squire et al., 1995; Montiel et al., 2018). Here we attribute the attenuation entirely to

the dissipative process with the following arguments. Ardhuin et al. (2018) reported that in the studied region,

ice floe scattering is a weaker effect on wave attenuation compared with other processes, including the

boundary layer beneath the ice, inelastic flexing of ice cover, and wave-induced ice fracture. The inelastic

flexing and ice fracture may be considered as part of the dissipative mechanism within the ice cover already

included in the viscous coefficient, but scattering is a re-distribution of energy, which must be isolated from

the apparent attenuation before using the data to calibrate the viscoelastic models. We estimate the scattering

effect based on the study of Bennetts and Squire (2012) as follows. In that study, wave attenuation by floes,

cracks and pressure ridges were examined. In the absence of in-situ observations, we assume that few and

small ridges are present in the studied ice cover (thickness < 0.3 m), hence the effect of ridges is negligible.

In our case of long waves propagating through such thin ice cover, Bennetts and Squire (2012) have shown

that the floes produce much more attenuation than the cracks. Without in-situ observation, WW3 simulations

(REF run in Ardhuin et al., 2018) implementing wave-induced fracturing of ice floes gave a range of

estimated maximum floe diameter from 70 to 150 m. Using this range of floe diameter, the theoretical

scattering results from Bennetts and Squire (2012) indicate that the floe scattering-induced attenuation is

about $10^{-7}$ m$^{-1}$, which is negligible compared to the $\alpha$ shown in Figure 3.

     With all the above simplifications and the assumed exponential decay of wave energy, Eq. (2) becomes:

$$-c_g 2\alpha E = (1 - C)(S_{in} + S_{ds}) + S_{nl} - C2c_g k_i E \qquad (3)$$

which yields

$$k_i = \frac{2c_g \alpha E + (1-C)(S_{in}+S_{ds}) + S_{nl}}{2Cc_g E} \qquad (4)$$

where $k_i$ is the attenuation rate due to the ice cover. The occurrence of the $k_i$ data is presented by two-dimensional histograms of $k_i$ against $k_r$ in Figure 5. Not surprisingly, since wind effect is low due to the low open water fraction in the study region, and the relatively short distances between the pairs for the nonlinear

transfer of energy to accumulate, we observe that $k_i$ is very close to the apparent attenuation $\alpha$. We will discuss further this ice-induced dissipation in section 5.

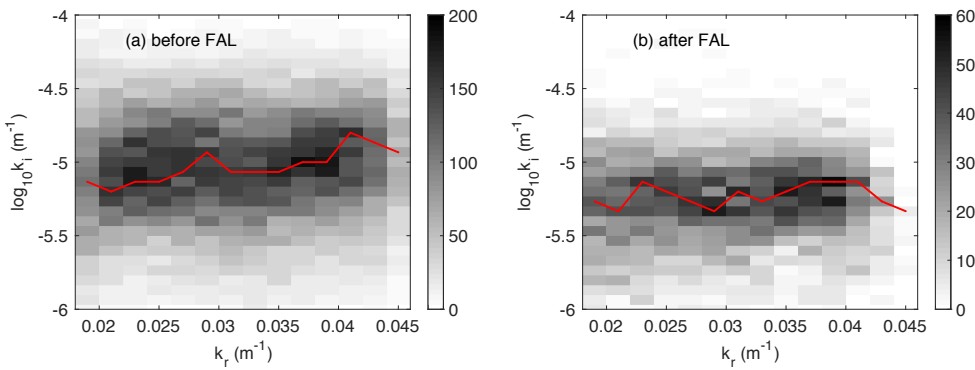

**Figure 5. Smoothed two-dimensional histograms of the ice-induced attenuation rate $k_i$ against wavenumber $k_r$, (a) before the FAL and (b) after the FAL. The attenuation domain is equally divided into 30 bins in the log scale**

**from $10^{-6}$ to $10^{-4}$ m$^{-1}$. Grayscale indicates the occurrence of $k_i$ in each $k_i$-$k_r$ bin from the selected pairs. The red curve indicates the highest occurrence of $k_i$ against $k_r$.**

## 4 Wave-in-ice model calibration

In this section, we present the calibration of the WS model and the FS model. We will begin with a brief introduction of the two models then describe the calibration procedure. In both models, $k_r$ and $k_i$ are solved

from the respective dispersion relations for given ice thickness and viscoelastic parameters. The viscoelastic parameters are optimized by minimizing the overall difference of $k_i - k_r$ relationship between the models and data obtained in section 3.

### 4.1 Dispersion relation

The dispersion relations of the two models are written as

$$\sigma^2 - Qgk \tanh kH = 0 \qquad (5a)$$

For the WS model,

$$Q = 1 + \frac{\rho_{ice}}{\rho_{water}} \frac{(g^2 k^2 - N^4 - 16k^6 a^2 v_e^4) S_k S_a - 8k^3 a v_e^2 N^2 (C_k C_a - 1)}{gk(4k^3 a v_e^2 S_k C_a + N^2 S_a C_k - gk S_k S_a)} \qquad (5b)$$

For the FS model,

$$Q = \frac{G_v h^3}{6\rho_{water} g}(1+V)k^4 - \frac{\rho_{ice} h \sigma^2}{\rho_{water} g} + 1 \qquad (5c)$$

where $H$ is water depth, $\sigma = 2\pi f$ is the angular frequency, $\rho_{ice}$ and $\rho_{water}$ are the densities of ice and water,

respectively, $k = k_r + ik_i$ is a complex wavenumber, $h$ is the ice thickness, $a^2 = k^2 - \frac{i\sigma}{\nu_e}$, $S_k = \sinh kh$,

$S_a = \sinh ah$, $C_k = \cosh kh$, $C_a = \cosh ah$, $N = \sigma + 2ik^2\nu_e$, $G_\nu = G - i\sigma\nu\rho_{ice}$ and $\nu_e = \nu + \frac{iG}{\rho_{ice}\sigma}$ , V is

the Poisson's ratio. Equivalent shear modulus $G$ and kinematic viscosity $\nu$ in both models are to be calibrated.

In this study, we use $H$ = 1000 m for deep water, $\rho_{water}$ = 1025 kg/m³, $\rho_{ice}$= 922.5 kg/m³ and V = 0.3 for

ice.

Hereafter, we use superscripts $t$ for the theoretical values and $m$ for the measured data. Specifically, the

theoretical wavenumber and attenuation rate are denoted as $k_r^t$ and $k_i^t$. They are calculated for each set of

$G, \nu$ values. Attenuation data due to the ice effect obtained in section 3 is denoted as $k_i^m$. The corresponding

wavenumber $k_r^m$ is the discretized $k_r$ values from 0.011 m⁻¹ to 0.0045 m⁻¹ with an increment of 0.002 m⁻¹.

**4.2 Calibration methodology**

In this section, we optimize $G, \nu$ by fitting $k_i - k_r$ relationship from the model via Eq. (5) to the measured

values from Eq. (4). Specifically, for given $G$ and $\nu$, we solve arrays of $k_r^t$ and $k_i^t$ through Eq. (5) for each $f$

densely sampled from 0.0001 Hz to 1 Hz. We then interpolate the results to obtain the theoretical $k_i$ at each

wavenumber $k_r^m$ from the data in section 3.2. For each of these $k_r^m$ we have the measured dissipation rate

$k_i^m$ shown in section 3.2. We now use an optimization procedure to determine the best-fit parameters $G$ and

$\nu$. The objective function for the optimization is defined as the weighted sum of the differences between $k_i^t$

and $k_i^m$ over $k_r^m$, i.e.,

$$F = \min_{G,\nu}\left\|w(k_r^m)\big(k_i^m(k_r^m) - k_i^t(k_r^m)\big)\right\|_2 \tag{6}$$

where $\|\cdot\|_2$ is the L-2 norm operator, $w(k_r^m)$ is a weighting factor to account for the distributions of wave

energy and attenuation rate. Cheng et al. (2017) tested two weighting factors: $w = \int E(\theta, f)d\theta$ and

$\int E(\theta, f)f^4 d\theta$ in calibrating the WS model. In that study, the measured data had a range of $f$ varied from

0.05 to 0.5 Hz, and the attenuation rate varied from 10⁻⁶ to 10⁻³ m²/s. The authors found that using $w = $

$\int E(\theta, f)d\theta$ fitted attenuation best at the most energetic wave band, while $w = \int E(\theta, f)f^4 d\theta$ performed

better to capture significant increasing of $k_i$ at high frequencies. No weighting factor could produce a fitting

over the entire spectral attenuation curve. For the present study, the variation of spectral attenuation is small

as shown in Figure 5 with no particular region of emphasis, we thus choose $w = \sqrt{\int E(\theta, k_r)d\theta}$ which has

a broad band around the peak energy of the wave field.

We choose a search domain (10⁻⁷ Pa $\leq G \leq$ 10¹⁰ Pa and 10⁻⁴ m²/s $\leq \nu \leq$ 10⁴ m²/s) for the WS model. This

search domain covers all other reported viscoelastic values for ice covers (e.g., Newyear and Martin, 1999;

Doble et al., 2015; Zhao and Shen, 2015; Rabault et al., 2017). For the FS model, it is known that very large

$G, \nu$ are needed to obtain the level of $k_i$ observed (Mosig et al., 2015). We thus choose a large search domain

10 Pa $\leq G \leq$ 10²⁰ Pa and 10 $\leq \nu \leq$ 10¹⁵ m²/s. This parameter range is far beyond the measured data from

solid ice (Weeks and Assur, 1967). The global optimization procedure is performed by using the genetic algorithm with function *ga* in MATLAB and Global Optimization Algorithm Toolbox (R2016a).

**4.3 Results**

For the WS model, the optimized $G, \nu$ are gathered into a small cluster around $G \cong 10^{-4}$ Pa and a large cluster around $G \cong 10^5$ Pa with large variation in $\nu$. The existence of two separate clusters of calibrated results was also found in grease/pancake mixtures near the ice edge (Cheng et al., 2017). Note that multiple solutions (optimized $G, \nu$ pairs) could be obtained in optimizing a nonlinear system such as shown in Eq. (5). Thus, constraints are applied to select solutions that are physically plausible. Note that the solutions around $G = 10^{-4}$ Pa, and solutions with $\nu$ near $10^4$ m²/s lead to the residual from Eq. (6) insensitive to $G$. It implies that the modeled material is viscous dominant with almost nil elastic effect. Those cases are incompatible with the pack ice condition, thus removed in the following analysis. Scatter plots of the remaining data points $(G, \nu)$ are presented by Figure S2 in the supplemental material. For these remaining data, we apply the bivariate Gaussian distribution to obtain the 90% probability range of $(G, \nu)$ before and after the FAL, respectively. Means and covariances of the related probability density functions are given in Table 1. A slight difference of distributions of $G, \nu$ before and after the FAL is noticed. The mean of elasticity (viscosity) is slightly lower (higher) before the FAL than that after the FAL. The results imply that the ice layer behaves more elastic in the inner ice field and more viscous towards the ice edge. It is worth noting that the ranges of both $G$ and $\nu$ are about one order of magnitude greater than those of the grease/pancake ice (Cheng et al., 2017). As expected, the effective elasticity in the WS model for more solid ice is closer to the elastic modulus of sea ice.

**Table 1. Statistical features of the clusters of $G$ and $\nu$ from the WS model and FS model calibration. The mean ($\mu$) and covariance (*cov*) of $X = \log_{10} G$ and $Y = \log_{10} \nu$**

|  | $\mu_X$ | $\mu_Y$ | $cov(X,X)$ | $cov(X,Y)$ | $cov(Y,Y)$ |
|---|---|---|---|---|---|
| WS Model |  |  |  |  |  |
| Before FAL | 5.07 | 1.51 | 0.07 | 0.07 | 0.81 |
| After FAL | 5.25 | 1.32 | 0.11 | 0.27 | 1.26 |
| FS Model |  |  |  |  |  |
| Before FAL | 17.39 | 10.82 | 2.83 | 1.42 | 1.00 |
| After FAL | 16.26 | 9.51 | 1.90 | 0.95 | 0.75 |

For the FS model, the calibrated $(G, \nu)$ are clustered, where scatter plots of $(G, \nu)$ are given in Figure S3 in the supplemental material. Hence all data points are used in the bivariate Gaussian fitting. The resulting mean and covariance values are also given in Table 1. Notice that the mean values of $G, \nu$ from the FS model are extremely larger than the intrinsic values of ice. The distribution of calibrated values is further discussed in the discussion section.

Hereafter we only elaborate on the results of the WS model. Figures 6(a) shows the overall comparison of $k_i - k_r$ between the measurements (gray) and the WS model (black). Both the median (solid curve) and 90% boundaries (dash curves) are in good agreement. In Figure 6(a), we superimpose an empirical model from Meylan et al. (2014), which is also included in WW3 to account for the ice effect as switch IC4. By fitting the wave buoy data from the Antarctic MIZ obtained in 2012, Meylan et al. (2014) proposed a simple period-

dependent attenuation rate $k_i(T) = \frac{2.12 \times 10^{-3}}{T^2} + \frac{4.59 \times 10^{-2}}{T^4}$, which is converted into a $k_i - k_r$ relation through the open water dispersion relation. The empirical formula predicted that attenuation was consistent with the range we obtained here, but with a higher sensitivity to $k_r$. Figure 6(b) shows the normalized wavenumber $\frac{k_r}{k_{ow}}$ against $f$ using the optimized $G, \nu$ in the WS model. The deviation of $\frac{k_r}{k_{ow}}$ from 1 is less than 5%, indicating the modeled wave dispersion agrees with the open water dispersion relation. It is consistent with

Monteban et al. (2019) who investigated wave dispersion in ice from the SAR data in the Barents Sea, as the studied range of wavelength and ice thickness are similar. The counterpart of the FS model is given Figure S4 in the supplemental material.

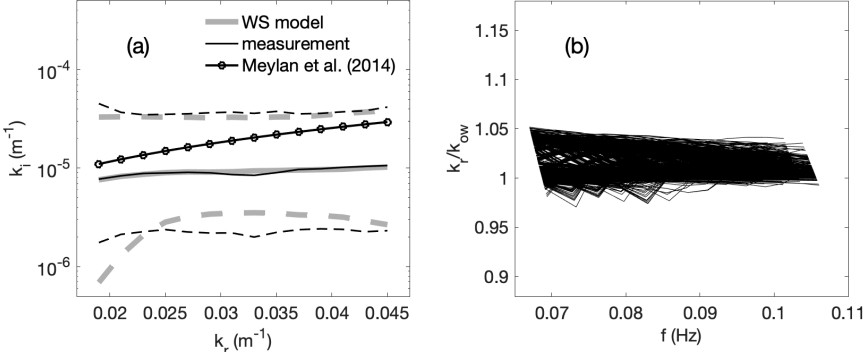

**Figure 6. (a) Comparison of ice-induced attenuation $k_i$ between measured data from Eq. (4) and the WS model**
**predictions; Solid lines are mean values and dashed lines are 90% confidence intervals. Gray-thick lines are the calibrated WS model and black thin lines from the data. Black line with symbol is the empirical model from Meylan et al. (2014); (b) the corresponding $k_r/k_{ow}$ from the calibrated WS model against wave frequency.**

Figures 7(a)(b) show distributions of calibrated $G$ and $\nu$ from the WS model in the azimuth-range domain, respectively. Contours represent ice thickness distribution from SMOS. The FAL is marked as red dots. The
spatial domain is divided into 12.5 km × 12.5 km cells to enhance visualization. Cell color indicates the averaged value of $\log_{10} G$ and $\log_{10} \nu$ of all pairs with midpoints inside the cell. Ignoring the outliers, Figure 7(a) shows that $G$ is generally larger after the FAL (thicker ice with leads) than before the FAL, while $\nu$ has an opposite trend. Note that the outliers could be from multiple sources, such as noise in the retrieved wave data, spatial variability of ice condition, assumptions made in selecting pairs and calculating attenuation, and
the adopted genetic algorithm in the model calibration. The counterpart of the FS model is given in Figure S5 in the supplemental material.

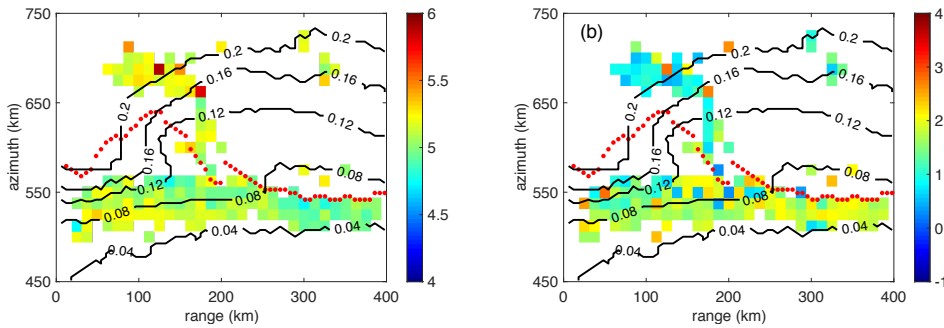

**Figure 7. (a) Distribution of $\log_{10} G$ in averaged 12.5 km×12.5 km grid in the range-azimuth plane. Cell color indicates the averaged $\log_{10} G$ of all pairs with midpoints inside a cell; Contours indicate SMOS ice thickness; Red dots indicate the FAL (b) Same as panel (a) except replacing $\log_{10} G$ with $\log_{10} \nu$.**

## 5 Discussion

In this section, we discuss some key wave characteristics mentioned in the analysis, and the behaviors of calibrated model parameters. Some thoughts about wave-in-ice modeling are provided at the end.

### 5.1 Evolution of wave characteristics

The behaviors of $k_{r,dominant}$ and PSD in Figures 1 and 4 may come from multiple mechanisms. The decrease of $k_{r,dominant}$ towards the interior ice is in agreement with a similar study in the MIZ (Shen et al., 2018), as well as the lengthening of dominant wave periods reported in many field observations (e.g., Robin, 1963; Wadhams et al., 1988; Marko, 2003; Kohout et al. 2014; Collins et al., 2015). The phenomenon is commonly explained by the low pass filter mechanism in literature. That is, higher frequency waves are preferentially attenuated, thus the power spectrum peak shifts to longer waves. However, though this mechanism is supported by the increasing trend of $k_i$ with increasing wavenumber before the FAL, it is insufficient to explain the observations after the FAL, where the $k_i - k_r$ curve is practically flat (Figure 5) and the peak of PSD shifts toward lower $k_r$ with increasing latitude (Figure 4). A downshift of wave energy towards lower wavenumber might also be related to nonlinear wave-wave interaction. Through which, spectral wave energy in the main wave frequency could transfer into the low-frequency components. How to quantify these mechanisms is still an open question with little data especially in pack ice, where the skills in observations are still developing. A piece of critical information is the relationship between the wave period and wave energy, especially in the region after the FAL. Monteban et al (2019) showed promising applications of overlapping SAR images separated by a sufficient time gap to obtain such information.

The attenuation rate obtained in this study ($\sim 10^{-5}$ m$^2$/s) against wavenumber shows a slightly increasing trend before the FAL, while nearly flat trend and lower after the FAL. Similar magnitude of attenuation is found in the MIZ in both Arctic and Antarctic as reported in previous studies in the same period range but larger for shorter periods (e.g., Meylan et al., 2014; Doble et al., 2015; Rogers et al., 2016; Cheng et al., 2017). The analysis of attenuation against wavenumber in section 3 is obtained for pairs of observations over relatively

short distances (<60 km). In Appendix B, we present another method to determine the overall attenuation, similar to the analysis of the attenuation of the significant wave height shown in Stopa et al. (2018b). By analyzing wavenumber by wavenumber over the entire space where the SAR data covers, the apparent attenuation coefficient is obtained by fitting hundreds of data over long distances, ignoring the higher variation of ice thickness (0.01~0.3 m) and the shift of dominant wavenumber from the PSD curves. The results are shown in Appendix B, Figure B2. In some cases, we can even have negative values of this attenuation, meaning energy increases as the wave propagates. What we would like to emphasize by showing this result is that over a very long-distance ice condition can change significantly, in addition to nonlinear energy transfer and wind input/dissipation, thus long-distance spectral analysis may become difficult to interpret.

## 5.2 About model calibration

The covariance values of the calibrated viscoelastic parameters are greater than those found in grease/pancake ice using buoy data (Cheng et al. 2017). In processing the SAR data, there are many challenges to be dealt with. It is extremely difficult to separate 1) sea ice variability, 2) wave height variability, and 3) instrument variability (speckle and in-coherent SAR noise). All of these influence the variations we see in the wave spectral data. Still, the scatter of the spectral attenuation shown in Figure 3 is significant even after the filtering described in section 3.1. This data scatter results in a large range of calibrated model parameters. We believe that the scatter of calibrated $G, \nu$ in both WS and FS models could be narrowed down by reducing the uncertainty of measured attenuation data. We also believe that for the FS model the calibrated values and spread could be reduced by modifying the objective function (Eq. (6)) with an additional constraint on the wavenumber, which presently shows an extremely large range as shown in Figure S4. Regardless, the FS model needs much larger $G, \nu$ to produce attenuation rates comparable to the observed data. This is the case not only for the pack ice but also for the MIZ. When analyzing data from the Antarctic MIZ reported in Kohout and Williams (2013), and by further constraining $k_r = k_{ow}/1.7$, Mosig et al. (2015) obtained $G = 4.9 \times 10^{12} \text{Pa}, \nu = 5 \times 10^7 \text{m}^2/\text{s}$. Though far below the present case, these values are still orders of magnitude above the intrinsic values measured from solid ice (Weeks and Assure, 1967). Meylan et al. (2018) discussed the attenuation behavior among different dissipative models by assuming small $|k_r - k_{ow}|$. Specifically, they showed that the FS model produced $k_i \approx \frac{\rho_{ice}(1+V)h^3}{6\rho_{water}g^6} \nu \sigma^{11}$ and the pure viscous case of WS model (i.e., Keller (1998)'s model) produced $k_i \approx \frac{4\rho_{ice}h}{\rho_{water}g^4} \nu \sigma^7$. The higher the power in $\sigma$, the higher $\nu$ is to match the measured $k_i$ in the high period (low frequency) waves range. The FS model also naturally leads to large $G$. As shown in Mosig et al. (2015), inverting the dispersion relation shown in Eqs. (5a,c) gives $G - i\sigma\rho_i \nu = 6\frac{\rho_w\sigma^2 - gk\rho_w - hk\rho_i\sigma^2}{h^3k^5(1+V)}$. The leading term of which in small $k_r$ yields $G \approx O(k_r^{-5})$.

**5.3 Thoughts on modeling wave-ice interaction**

Damping models play a crucial role in spectral attenuation. At present, to use any specific model to describe wave attenuation is tentative. The identified damping mechanisms are many. For instance, boundary layer under the ice cover (Liu and Mollo-Christensen, 1988; Smith and Thomson, 2019), spilling of water over the ice cover and interactions between floes (Bennetts and Williams, 2015; Herman et al., 2019a, b), jet formation between colliding floes (Rabault, 2019), have all been reported in laboratory or field studies. A full waves-in-ice model that considers all important mechanisms is not yet available. While theoretical improvements are needed to better model wave propagation through various types of ice covers, practical applications cannot wait. Calibrated models that are capable of reproducing key observations must be developed in parallel to model improvements. The present study provides a viable way to calibrate two such models available in WW3. These models lump all dissipative mechanisms in the ice cover into a viscous term. This type of model calibration studies has two obvious utilities. One, with proper calibration, models can capture the attenuation of the most energetic part of the wave spectrum. Two, the discrepancies may be used to motivate model development that includes missing mechanisms, thereby help future model development. Because different physical processes may play different roles under various ice morphology, collecting wave data to calibrate these models under various ice types is necessary. Finally, more observations with higher quality data will improve the modeling of the wave-ice interaction and the robustness of the models.

**6 Conclusions**

In this study, we use the wave spectra retrieved from SAR imagery to examine the spectral attenuation in young pack ice. These images were obtained in the Beaufort Sea on 12 October 2015. According to the analysis of data retrieved over several hundred kilometers, the observed decrease of wave energy and lengthening of dominant waves towards the interior ice are consistent with earlier in-situ observations. We investigate wave attenuation of dominant spectral densities per wavenumber between two arbitrary locations in the region separated by less than 60 km. Similar attenuation rates are observed for all wavenumbers from 0.019 to 0.045 $m^{-1}$ (estimated wave period from 9 to 15 s). After isolating the ice induced attenuation out from these data, we calibrate two viscoelastic-type wave-in-ice models through an optimization procedure between the measured data and theoretical results. Both models can generally match the observed attenuation corresponding to the energetic portion of the wave spectra, with a large difference in dispersion (see wavenumber plots in Figure 6 and Figure S4). For the WS model, the calibrated shear modulus (viscous parameter) in the region beyond the first appearance of leads with thicker ice is slightly larger (smaller) than that of the region closer to the ice edge before the first appearance of leads. The resulting wavenumber is within 5% of that from the open water dispersion.

Wave-ice interaction is complicated due to many co-existing physical processes. At present no models have fully integrated all identified processes. The present study demonstrates a method based on measured data to calibrate existing models so that they can be applied to meet the operational needs. As the models improve,

further calibration exercises may be performed accordingly. For example, the eddy viscosity model (Liu and Mollo-Christensen, 1988) attributed the wave attenuation entirely to the water body under the ice cover. This model is also included in WW3 as the switch IC2, with the eddy viscosity as a tuning parameter. The present

analysis can also be used to calibrate that model or models that try to combine both dissipation from the ice cover and the eddy viscosity from the water beneath (e.g. Zhao and Shen, 2018). From a physical point of view, dissipative mechanisms may be present simultaneously inside the ice cover and the water body underneath. However, the calibration of complex models with multiple co-existing processes is a difficult task that requires a much more dedicated study.

We conclude by noting that high-resolution spatial data from remote sensing provide new opportunities to investigate the wave-ice interaction over a large distance and different ice types. However, details of ice types and temporal observations are in development. To reach a full understanding and thus a complete waves-in-ice model requires collaboration from observation and modeling efforts.

**Appendix A**

The definition of the first appearance of leads was introduced in Stopa et al. (2018b). Here we provide more details of the methodology used. The SAR sea surface roughness images in Figure 1 of Stopa et al. (2018b) are divided into 5.1x7.2 km subimages with a 50% overlap of adjacent subimages in the range-azimuth domain. Each subimage contains 512×512 pixels. The FAL location for each range-position is defined as the minimum azimuth position where large-scale ice features were detected. Detection of large-scale ice features

is applied to each SAR subimage as the following. We first compute a one-dimensional spectrum of the SAR subimage to produce an image modulation spectrum. The spectrum is then normalized by the maximum energy contained in wavelengths from 100 to 300 m (the wavelength range of the dominant sea state for this event). When the ratio of the average of the normalized image spectra with wavelengths in the range of 600-1000 m and the dominant ocean-wave wavelength range from 160-220 m exceeds 0.8, we deem that there is

a "large-scale" feature such as lead within the image. Figure A1 shows two representative examples of detecting ice leads from SAR images captured before and after the FAL. From the criterion above, there are no leads in the top case, but leads are found in the bottom case. Also notice the change in the probability distribution of the roughness: the mean value changes (lower in the non-lead case compared lead case) and the standard deviation (lower in the non-lead case compared to the lead case).

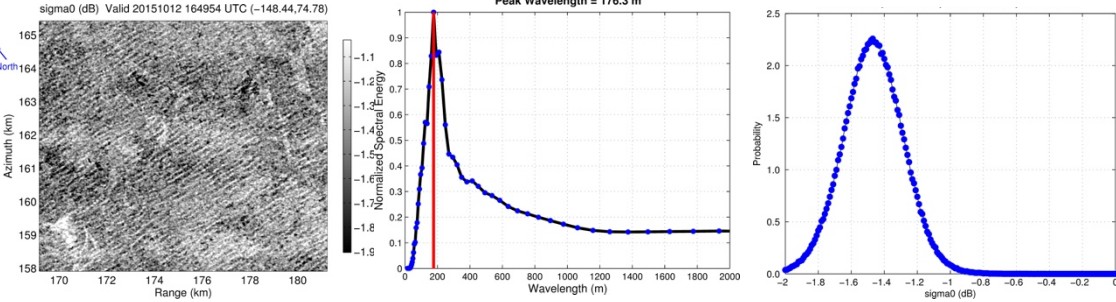


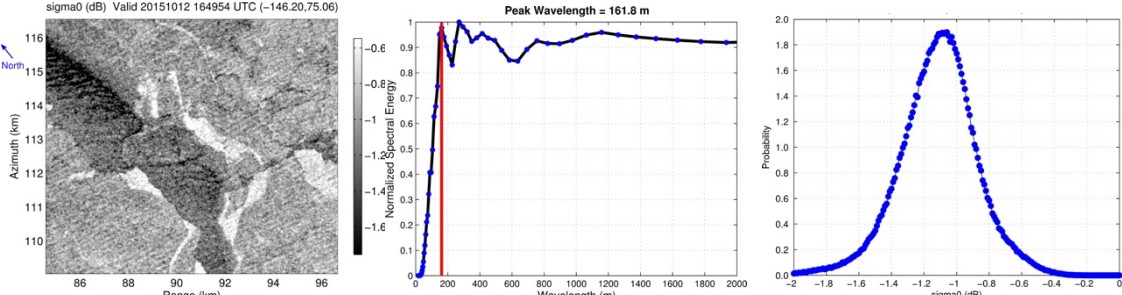

Figure A1. Illustration of the process to determine the FAL using two representative SAR subimages. (left) Surface SAR subimage roughness for a case located before the FAL (top) and a case located after the FAL.

(middle) Normalized spectral energy (normalized by the maximum energy within the 100-300 m wavelengths) of the SAR subimages where the red line indicates the dominant wavelength. (right) The probability density function of the SAR roughness (backscatter or sigma0 of thermal noise in the SAR imagery) for the two cases.

**Appendix B**

We investigate the decay of the dominant energy component $E(k_r, \theta_{k_r})$ over a long distance along selected tracks with fixed longitude (145°W, 146°W, …, 150°W). Figure B1 shows $E(k_r, \theta_{k_r})$ (blue dots) collected within a given longitude interval $150 \pm 0.1$°W starting from 74.5°N towards the north. $E(k_r)$ for $k_r < 0.019$ m⁻¹ is too scattered to show any attenuation trend. While as $k_r$ increases, the data become tighter with a clear decay trend. To obtain attenuation coefficient $\hat{\alpha}(k_r)$, we fit the $E(k_r, \theta_{k_r})$ by an exponential curve (black

line) in each panel based on an exponential wave decay assumption.

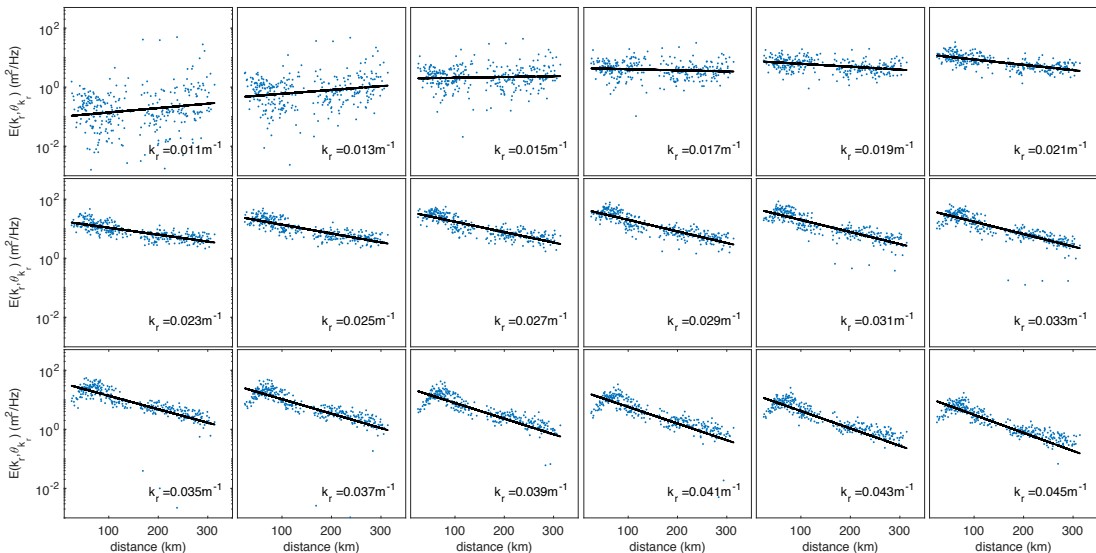

**Figure B1. Evolution of dominant wave energy component over long distances at different wavenumbers. The horizontal axis is the distance along a longitude from 74.5°N towards the north.**

The resulting $\hat{\alpha}(k_r)$ against $k_r$ is shown in Figure B2, as well as another five curves associated with different longitudes, processed in the same way. Figure B2 shows an increasing trend of $\hat{\alpha}(k_r)$ against $k_r$ as expected, and $\hat{\alpha}(k_r < 0.019 \text{ m}^{-1})$ are mostly negative. This method masks the variation of ice morphology and peak shifting of the power spectral density, thus it is insufficient to understand the damping mechanism in water-ice interaction. Hence, as our interest is to determine the attenuation in more consistent ice conditions, shorter distance between measuring points is adopted in section 3.

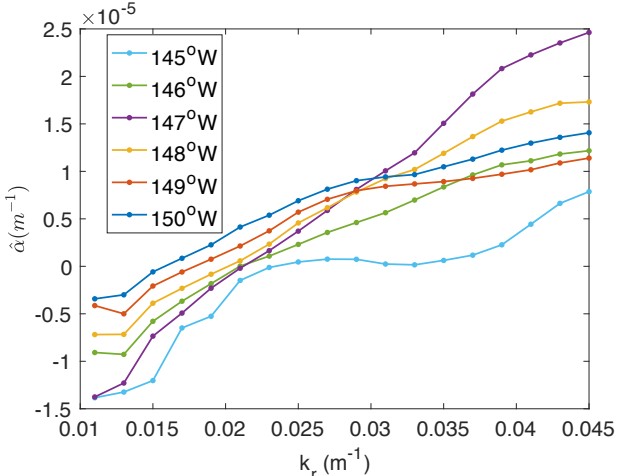

**Figure B2. $\hat{\alpha}$ against $k_r$ of different longitude tracks.**

**Code and data availability**

The code and data necessary to reproduce the results presented in this paper can be obtained from the corresponding author.

**Author contribution**

Stopa and Ardhuin retrieved the wave spectra dataset from SAR images. Cheng and Shen planned the research and prepared the manuscript with contributions from all authors. Cheng performed the analysis.

**Competing interests**

The authors declare that they have no conflict of interest.

**Acknowledgements**

The present work is supported by EU-FP7 project SWARP under grant agreement 607476, Office of Naval Research grant numbers N000141310294, N000141712862, and N0001416WX01117. Data, and a cruise

report can be obtained at:https://drive.google.com/open?id=0B9Au2ZqQ-BM5YTJPWXBsV2Q0THc. The raw Sentinel-1A SAR data are provided by Copernicus and are available on-line via the Open Access Data Hub (https://scihub.copernicus.eu/). AMSR2 sea ice concentrations are from http://doi.org/10.5067/AMSR2/A2_SI12_NRT. SMOS sea ice thickness data are from https://icdc.cen.uni-hamburg.de/1/daten/cryo-sphere/l3c-smos-sit.html. CFSR wind data are from https://rda.ucar.edu/datasets/ds094.0/. OSCAR Current data are from https://www.esr.org/research/oscar/.

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
