# Peer review of "Spectral attenuation of ocean waves in pack ice and its application in calibrating viscoelastic wave-in-ice models"

_The Cryosphere, 2019_

## Referee Comment (RC1) · Anonymous Referee #1 · 16 Jan 2020

In their manuscript „Spectral attenuation of gravity wave and model calibration in pack ice", Sukun Cheng and colleagues present results of an analysis of wave energy attenuation based on data obtained from a set of SAR scenes from the Beaufort Sea. The analysis includes: (i) derivation of spectral wave characteristics in the area of interest, divided into two sub-regions with different ice types and morphology, (ii) computation of linear attenuation coefficients for a large number of pairs of points located in both sub-regions, and (iii) calibration of parameters of two selected models of wave attenuation, by Fox and Squire, and Wang and Shen, to the observed spectral attenuation. The manuscript also includes a discussion of possible sources of errors in the analysis, data deficiencies, as well as a more general discussion of problems with model calibration related to a large number of unknown coefficients and with the fact that a multitude of different physical mechanisms contribute to the net attenuation observed in the field.

It is relatively easy to point out limitations of this type of analysis, but – as the Authors rightfully remark – our limited understanding of the processes involved, combined with limited availability of data for model calibration and validation, restrict our ability to develop complex, physics-based models and justify development of simplified, but practically applicable parameterizations (like those implemented in the WW3 wave model). Therefore, in my opinion, the work presented in the manuscript is very valuable and has several aspects practically relevant for spectral modeling of wave propagation and dissipation in sea ice. I think that the results are worth publishing in "The Cryosphere". My comments on the manuscript are listed below.

General comments:

1. The text of the manuscript contains a lot of (mostly small) grammar, punctuation and other language mistakes and should be carefully corrected before publication.
2. I'd suggest modifying the title of the paper. I understand the Authors wanted the title to be short, but in my opinion they overdid it. "Model calibration in pack ice" – what kind of a model? It might mean anything. I'd also suggest changing "gravity wave" to "gravity waves".
3. The location of FAL – and its very existence – is crucial to the analysis presented in this paper. The Authors first introduce this term on page 3 (lines 75-76), suggesting that it was used (or defined) by Stopa et al. (2018b). It should be Stopa et al. (2018a) – see also my technical comment no. 1 below. But, more importantly, even if that information is provided in the previous papers, I'd suggest adding it to the present manuscript as well: how was the position of FAL determined? How does the ice cover differ on both sides of the FAL-line? In the present form, the FAL seems rather "mysterious". For example, further on page 3 we read: "…the FAL (black dots) presumably marks the separation between discrete floes and a semi-continuous ice cover with dispersed leads". (A bit further, in line 120, again: "presumably a semi-continuous cover".) Presumably? Does it mean those features cannot be unambigously identified in the analyzed images? How then was the position of FAL determined? What was the criterion? What is the uncertainty associated with the location of FAL? Very importantly: was the location of FAL determined independently of any information on wave characteristics?
Could the authors add a figure showing fragments of the analyzed images on both sides of FAL (not necessarily in the main text, but in the supplement)?
To make it clear: I'm not criticizing the analysis nor the way FAL was defined/identified, but the presentation in the manuscript.
4. As far as I know from other studies (I'm not an expert in satellite data analysis), the satellite algorithms used to determine ice concentration and thickness perform relatively poor in thin,

"new" ice types (frazil, grease, pancake ice). Could the Authors comment on the reliability of the concentration and thickness maps (Fig. 1b,c) in the region south of FAL, where the thickness is 10 cm or less? Is the apparent west-east gradient of ice concentration and thickness in that region really present or is it possible that in fact it is a change of ice type? Those questions are important for some aspects of the analysis, for example, in line 118, where the Authors say that the wavenumber "varies with ice concentration but is insensitive to ice thickness variation...".

Other, mostly technical comments:

1. Lines 69-70: "… (refer to Figure 1 in Stopa et al. (2018b))". It should be 2018a, shouldn't it? 2018b is about the Amundsen Sea, not the Arctic.
   I'd suggest checking the references to those two papers throughout the text.
2. Line 74: contaminated -> contamination
3. Line 94: "This work is to…" Change to "The purpose/goal of this work is…"
4. Line 95: "(<0.3 m)" – I'd suggest changing to "(thickness<0.3 m)" (and similarly in line 247). Further in that line: "dominant region". In which sense is that region "dominant"?
5. Figure 1: panels (b) and (c) are marked, but "(a)" is missing from panel (a).
6. Line 165: "… based on the sensitivity of the number of selected pairs depending on these values". I don't understand this sentence.
7. Line 167: $10^{-4}$
8. Lines 188/189: correspond -> corresponding; indicate -> indicating
9. Line 205 (caption of Fig. 4): Geological? You mean geographical?
10. Line 217: As Eq. (2) is time-dependent, it should be $E(k_r,\vartheta,x,t)$
11. Lines 231-233: It is worth mentioning that those parameterizations were formulated for open water.
12. Line 258: histograms of $k_i^m$, not $\alpha$
13. Line 266: "In both models, $k_r$, $k_i$…" It would be nice to introduce those variables (as well as $k=k_r+ik_i$) much earlier, because they (especially $k_r$) are used from the beginning.
14. Lines 363-365: The low-pass filtering mechanisms described here has a strong influence on the *average* wave period (and length), less so on the dominant period – unless attenuation takes place around the peak of the spectrum.
15. Line 412: "Thoughts of…" -> "Thoughts on…"

---

## Referee Comment (RC2) · Harry Heorton (Referee) · 17 Jan 2020

Spectral attenuation of gravity wave and model calibration in pack ice
Cheng et al
tc-2019-290

This paper documents a study of the attenuation of sea surface gravity waves by sea ice. The study involves the use of Sentinel 1A SAR imagery to extract first a wave number direction spectrum within a region of thin sea ice in the Beaufort Sea. Physical laws are introduced that use the SAR data to get the apparent wave attenuation and wave attenuation due to ice effect. A calibration scheme is then introduced that takes the modeled wave frequency spectrum, generates the wave attenuation due to sea ice, and then compares these values to the data. The calibration scheme is used to find optimal model parameters of the shear modulus and molecular viscosity to best reproduce the observed conditions. Optimal values are presented along with a detailed discussion of how they compare to previous studies. The detailed analysis of wave attenuation required to calibrate the wave model has then also been used to discuss the physics in play. A particular emphasis here, and through the analysis is on differing characteristics within the ice pack before or after lead formation. The authors do well to show the role of their work within the science of the highly complex system of wave ice interaction and the benefits of using the analysis and calibration presented. After the authors address points i have below regarding the documentation of the papers methodology I suggest it is fit for publication.

Whilst this paper well documents the technical parts of data analysis and model calibration, and throughly discusses the success, limitations and implications of their results, it is severely lacking in any general description of the methods used. I had to read the entire paper, and all the specific details of the methods before I could understand the general incentives and methods of the study. I have suggestions below for how to improve the paper to address this. Once I had worked out what methods of the paper were the various sections of the paper fell into place and the science behind the paper was well founded and thoroughly discussed. However, I still have some key questions.

1) The paper does not clearly explain the reader the format of the data used and the extent of analysis performed for this study. Do you take the wave energy spectrum from Stope 2018b or the wave number direction spectrum? Or you start with the raw SAR imagery and repeat the same analysis as before? I spent much of section 2 asking myself this. Also the format of the introduction adds to this confusion, whilst it starts with 4 paragraphs of overview, paragraph 5 is neither an introduction to the area of interest, or a description of the data used in this paper. The rest of the paper has a similar ambiguity between the the data sources for this study, the data processed as part of this paper and previous studies that are being cited for comparison.

2) There is little description within the paper of what is being documented by each section. The brief outline on lines 103-109 is again ambiguous to what is being performed. For example 'site description and wave characteristics are depicted in section 2' does not inform me as to whether section 2 contains description of the data used in the study and how you processed it, a summary of previous studies of the region and what they found, or even a description of a model run of the region. Section 2 itself starts immediately with a definition of $k_r$, which is clearly done, but there is no description at all of whether $k_r$ in this study is an original data source collected by the study or a previously collected data that is being analyzed here. This lack of overview is present in near all of the technical sections with the paper. I suggest adding a clear paper overview of the form; data was obtained from (here, here, etc) that we analyze thusly to extract this information. We then ran these models and using this scheme and the previously documented data we calibrated these parameters. This information also needs to be clearly defined at the beginning of each section. Once I had worked out this information for myself I was easily able to understand the technical parts of the study, which are well written.

3) There are two key values used in this study, apparent wave attenuation and the wave attenuation due to ice effect. The notation for these needs to be changed. $k_r$ is a wave number spectrum, alpha is the associated wave attenuation, but $k_i$ is just a wave attenuation whilst previous notation leads me to think it will also be a wave number. While the methods used to get these values are documented, there is no physical description of what these values represent and why you are interested in them. A paragraph in the introduction introducing these physical values, and the other values considered in this paper (dominant wave number, wave energy spectrum for example) will greatly help a non-specialist reader. Section 3 needs an introduction explaining the differences between kr and ki, and as mentioned before, the data you use to obtain these values. Also the wave attenuation due to ice effect ki and ki^m need to have clearer definitions. I see that you have used ki as the modeled attenuation, and ki^m as that data derived value. This notation needs adjusting, the use of superscript m to define the value that is not modeled is confusing. Section 3.2 will benefit from a definition of ki at the beginning, as currently the reader has to wait till the end of the section to understand what the aim of the presented method is.

 4) I suggest that the introduction section needs restructuring and expanding upon. Currently there is one paragraph on the place of wave sea-ice interaction within climate science and then a very technical description of one wave model. The paper would benefit from having first the next paragraph that explains the uses of wave forecast models, and then the technical description of WaveWatch follow. The final paragraph of the introduction needs to moved to a dedicated data description and expanded upon. It is currently unclear what SAR data used in this study is new and what has been previously published. If the SAR observational data and analysis or processing has been previously published, then a clear section citing this publication and describing the data is needed. If there is novel data work in this study then this needs to be clearly stated and the technique and data clearly described.

Specific edits:

'wave propagation' this is the first sentence in the paper. Please be more explicit in which waves you re discussing. Consider an additional sentence introducing ocean surface gravity waves
'fall 2015' pleas give a more accurate time. The study is for a single day i believe.
- 24 this sentence is very long. I had to re-read several times to understand what became lower where. Consider splitting.

These citations are 10+ years old now, the reduction and predictions have changed a lot in these 10 years.

Wavewatch is first introduced with a very technical description. Can you first give an overview of Wavewatch? Its uses and the incentives behind its creation. A brief description of its main physical constructs will also be useful to the reader. The ice effects as modeled by WaveWatch are described here, but I'd like to see how these ice effects sit within the whole model.

An introduction into how wave forecast models work would be useful here. I'd like to know how to use a wave forecast model in general, then I can better understand the context of your work in the wave sea-ice interaction aspect.
Here and above you mention IC3 (and IC5). You mention that they both store and dissipate mechanical energy but what are the differences between them.

This next paragraph needs moving to a dedicated data section. Within this section it needs to be made clearer what data was used as part of this study and why. Also I find it difficult to tell from this paragraph which data was collected together in a previous study and which data was brought together for this study. Those which were brought together here needed to be expanded upon. The benefits and limitations of each data need to be better discussed. Currently the novel method of Ardhuin 2017 and Stopa 2018a is well described, as are the reasons for its use in this paper. However there is less description for the Buoy and ice concentration data. Due to the data all being plotted together in figure 1, I am assuming that this combination of data sources is novel for this study. If this is the case please state so, and explain why the data have been chosen. If however these data have been combined before (which then explains why they are described in the introduction) then citations/descriptions of findings are needed.

Figure 1 There is no (a) label in the figure. Can you show on pane (a) how (b and c) colocate?

What is the methodology of calibrating IC3? Is it used extensively in this paper? If it is then a summary of the equational from is needed. If this is complicated then consider an appendix for the equations. If it is still too complicated then a paragraph explaining how it works with relevant citations is needed.

Next: a full description of wave energy spectrum $E(f,theta)$ is needed. Either the equational form of $E()$ needs to be added, or how it is produced by the model and how it is extracted from the data set. The source of the various $E()$ used in this paper is a current theoretical hole for me when reading it. As it is a key value for methods of this paper, consider a paragraph in the introduction explaining how the energy spectra is calculated from observations and treated within the model you are calibrating. Some background science on ocean surface gravity waves are described, including what the wave energy spectrum is would be a useful inclusion for this paper.

Also: At this point in the paper I don't want to have to dig into section 4 to discover what modification you have had to make to the methodology of Cheng 2017. At this point I flipped to section 4 and searched for both the 'Cheng' citation and the string 'IC3' within the paper and it was still unclear to me how the methodology of Cheng 2017 worked, why you had to modify it and what those modifications are. A description of all these aspects of this study is needed, either here, or at the beginning of section 4

I started reading this paragraph hoping to understand the incentives and aims of this study and what methods were used. The paper is unlikely to have been written in a linear fashion, but please remember that the reader will read it in a linear fashion, and is likely to be unaware of all the previous work on this topic, the data you are using, the model you are calibrating and the techniques you are using to calibrate it.

I am assuming that this section is a description of the data used in the study. However you give no indication of this in the text. I start by asking 'the wave number spectrum retrieved from where?'. This does not get answered, so the following information is very difficult to understand.
This range suggests you are talking about figure 1? I should have to work this out myself.
fitted to which $E(f,theta)$ from where?
I can't understand the direction sum of $E(kr,theta)$ when you have not described where $kr$ comes from and what $E(f,theta)$ is and where that comes from. Was there a previous analysis performed to get $E(f,theta)$? Was that done as part of this study or previously?
Again I am asking whether this work of extracting $kr,dominant$ is from this study. Was it previously done as part of Stopa 2018a/b ? Please inform the reader what work was done with the data for this study.
Difficult to see the variation with concentration in the figure.
Again difficult to see the decrease with $kr$ on the figure. Also it is unclear what you are referring to with 'wave propagation direction'.
what is the meteorological convention?
126. Please give a value here for $lambda\_c$ and/or the associated limit for $kr$.
Is the frequency $f$ estimated for all $kr$?

Section 3
Again this section really needs an introduction. I had to read the whole paper before I could make sense of it. Even so I still have some major questions that are easily answered by reading the paper.

please define the 'Pearson correlation coefficient'.
Power spectral density of what? I see this is mentioned in the figure, but please also define it in the text.

Figure 4. It would be nice to have an overlay square in figure 1 showing how the left column of figure 4 aligns. Please also note in the caption that the centre column is generated by pairs of data, whilst the rightmost comes from individual points.

224-229 I suggest that a version of this paragraph come at the beginning of the section. This would much ease the reading of this paper. Here I can understand clearly which values you are obtaining from where, why you chose them and what you intend to do with them. However the picture is still not entirely clear. This style of explanation is needed for every section and sub section.

and figures 3 and 5. A better description of these two figure and what you wish to show with them is needed. The caption for figure 5 is not sufficient. Saying that alpha vs kr is similar to ki^m vs kr is misleading, as alpha and ki^m are not similar physical quantities.

Here you say that kr and ki re solved for, but elsewhere in the paper kr is given as a data source whereas ki is a model variable. Is kr now referring to something else?

k = kr + i ki, how are you combining a wave number from data and an attenuation factor?
is the alpha here the same as in equation 1? Are you using alpha from the SAR data source in these equations to calibrate the model? If yes, then clearly say so, if not, then an alternative notation is required.
This sentence is an important one for this paper and needs to be clearly stated in the introduction.
This information would be better suited at the beginning of section 4 , before equations 5.
are you actually running WaveWatch as part of the calibration scheme? If so which out put variable are you taking for the calibration? I am assuming that you take the model out put frequency spectrum f, and use the relevant equations from 5 to create the required ki kr.
You have now introduced kr^m. What is this symbol for. The m would indicate 'model' but the opposite is true for ki. However previously kr was from a data source. Please clear up all the ki kr notation. Avoid using ^m if you're not going to use it to indicate 'model'
Equation 6 Is the subscript 2 supposed to be an exponent? If not what function does it refer to?

I would like to see a figure of the clustering of retrieved parameters. The table plus description does not give enough information.
311-312 Sentence unclear please revise
Ah I see the scatter plots are in the supplementary material. They would be a worthwhile inclusion in the main paper
Table 1. Please expand the caption to say what these numbers are. They are the results of the model calibration I assume?
Again put the scatter plots here, they are interesting.
Figure 6. Pane b illegible and either needs redrawing in color, omitting 90% of the lines, or even not including in the paper. Caption you mention 'the data' plotted in plot a, which data? from where? section 3 I guess.
It is important to also mention here the limitations of using a numerical (or a more specific description of the algorithm used) parameter search/optimisation scheme. For example: no. minima for equation 6 could be found for the data at these locations.
I find the discussion of ki here confusing due to the mixed notation used. Why not ki^m? Also here you talk about ice effect wave attenuation, but earlier you talk about kr,dominant which is a wave number. Please can you clear up the notation for these.
You here introduce a data mask, though this was not previously mentioned. Does it relate to the previous sentence on data quality? In which step of the data processing was this mask used?
Which function are you talking about? Equation reference?

Can you point me back to the results to show the difference in dispersion.

Appendix
I suggest moving the appendix to the main paper section. The results here appear to be as worthwhile as others discussed. Are these results from another study and you include them in an appendix for reference?

---

## Author Comment (AC1) · 26 Feb 2020

Response to Reviewer 1

Spectral attenuation of gravity wave and model calibration in pack ice Cheng et al. tc-2019-290

**Authors' response:**

We appreciate the reviewer's careful reading and the support of this study very much. A revised manuscript has not been prepared at the time due to the editor's request. We will revise the manuscript to respond to all of the issues raised by the reviewer. The review comments are listed below in black, and our responses are in red.

In their manuscript "Spectral attenuation of gravity wave and model calibration in pack ice", Sukun Cheng and colleagues present results of an analysis of wave energy attenuation based on data obtained from a set of SAR scenes from the Beaufort Sea. The analysis includes: (i) derivation of spectral wave characteristics in the area of interest, divided into two sub-regions with different ice types and morphology, (ii) computation of linear attenuation coefficients for a large number of pairs of points located in both sub-regions, and (iii) calibration of parameters of two selected models of wave attenuation, by Fox and Squire, and Wang and Shen, to the observed spectral attenuation. The manuscript also includes a discussion of possible sources of errors in the analysis, data deficiencies, as well as a more general discussion of problems with model calibration related to a large number of unknown coefficients and with the fact that a multitude of different physical mechanisms contribute to the net attenuation observed in the field. It is relatively easy to point out limitations of this type of analysis, but – as the Authors rightfully remark - our limited understanding of the processes involved, combined with limited availability of data for model calibration and validation, restrict our ability to develop complex, physicsbased models and justify development of simplified, but practically applicable parameterizations (like those implemented in the WW3 wave model). Therefore, in my opinion, the work presented in the manuscript is very valuable and has several aspects practically relevant for spectral modeling of wave propagation and dissipation in sea ice. I think that the results are worth publishing in "The Cryosphere". My comments on the manuscript are listed below.

General comments:

- The text of the manuscript contains a lot of (mostly small) grammar, punctuation and other language mistakes and should be carefully corrected before publication. We will clean up the language mistakes.
- 2. I'd suggest modifying the title of the paper. I understand the Authors wanted the title to be short, but in my opinion they over did it. "Model calibration in pack ice" what kind of a model? It might mean anything. I'd also suggest changing "gravity wave" to "gravity waves".

The title is revised as "Spectral attenuation of ocean waves in pack ice and its application in calibrating viscoelastic wave-in-ice models"

3. The location of FAL – and its very existence – is crucial to the analysis presented in this paper. The Authors first introduce this term on page 3 (lines 75-76), suggesting that it was used (or defined) by Stopa et al. (2018b). It should be Stopa et al. (2018a) – see also my technical comment no. 1 below. But, more importantly, even if that information is provided in the previous papers, I'd suggest adding it to the present manuscript as well: how was the position of FAL determined? How does the ice cover differ on both sides of the FAL-line? In the present form, the FAL seems rather "mysterious". For example, further on page 3 we read: "…the FAL (black dots) presumably marks the separation between discrete floes and a semi-continuous ice cover with dispersed leads". (A bit further, in line 120, again: "presumably a semi-continuous cover".) Presumably? Does it mean those features cannot be unambigously identified in the analyzed images? How then was the position of FAL determined? What was the criterion? What is the uncertainty

associated with the location of FAL? Very importantly: was the location of FAL determined independently of any information on wave characteristics? Could the authors add a figure showing fragments of the analyzed images on both sides of FAL (not necessarily in the main text, but in the supplement)?To make it clear: I'm not criticizing the analysis nor the way FAL was defined/identified, but the presentation in the manuscript.

We will include the explanation of the first appearance of leads (FAL) in Appendix A. "Appendix A

The definition of the first appearance of leads was introduced in Stopa et al. (2018b). Here we provide more details of the methodology used. The SAR sea surface roughness imagery in Figure 1 of Stopa et al. (2018b) are divided into 5.1x7.2 km subimages with a 50% overlap of adjacent subimages in the range-azimuth domain. Each subimage contains 512×512 pixels. The FAL location for each range-position is defined as the minimum azimuth position where large-scale ice features were detected. A detection of large-scale ice features is applied on each SAR subimage as the following. We first compute a one-dimensional spectrum of the SAR subimage to produce an image modulation spectrum. The spectrum is then normalized by the maximum energy contained in wavelengths from 100 to 300 m (the wavelength range of the dominant sea state for this event). When the ratio of the average of the normalized image spectra with wavelengths in the range of 600-1000 m and the dominant ocean-wave wavelength range from 160-220 m exceeds 0.8, we deem that there is a "large-scale" feature such as lead within the image. Figure A1 shows two representative examples of detecting ice leads from SAR images captured before and after the FAL. From the criterion above, there is no leads in the top case, but leads are found in the bottom case. Also notice the change in the probability distribution of the roughness: the mean value changes (lower in the nonlead case compared to the lead case) and the standard deviation (lower in the non-lead case compared to the lead case).

Figure A1. Illustration of the process to determine the FAL using two representative SAR subimages. (left) Surface SAR subimage roughness for a case located before the FAL (top) and a case located after the FAL. (middle) Normalized spectral energy (normalized by the maximum energy within the 100-300 m wavelengths) of the SAR subimages where the red line indicates the dominate wavelength. (right) The probability density function of the SAR roughness (backscatter or sigma0 of thermal noise in the SAR imagery) for the two cases."

As far as I know from other studies (I'm not an expert in satellite data analysis), the satellite algorithms used to determine ice concentration and thickness perform relatively poor in thin, "new" ice types (frazil, grease, pancake ice).

Could the Authors comment on the reliability of the concentration and thickness maps (Fig. 1b,c) in the region south of FAL, where the thickness is 10 cm or less? Is the apparent west-east gradient of ice concentration and thickness in that region really present or is it possible that in fact it is a change of ice type? Those questions are important for some aspects of the analysis, for example, in line 118, where the Authors say that the wavenumber "varies with ice concentration but is insensitive to ice thickness variation...".

To explain the use of AMSR2 and SMOS, we will add the following comments

"As shown in Cheng et al. (2017) (Supporting information Figures S6 and S7), these two ice products compared the best with in-situ observations in the MIZ. Their accuracies in the pack ice zone are uncertain."

---

## Author Comment (AC2) · 26 Feb 2020

Response to Reviewer 2

Spectral attenuation of gravity wave and model calibration in pack ice Cheng et al.
tc-2019-290

**Authors' response:**
We appreciative very much the careful reading of the reviewer. Whose suggestions should make this manuscript much more readable to a broader range of audience in polar science. A revised manuscript has not been prepared at the time due to the editor's request. We will revise the manuscript to respond to all of the issues raised by the reviewer. Our replies (in red) to the reviewer are added under each of the specific comments (in black).

This paper documents a study of the attenuation of sea surface gravity waves by sea ice. The study involves the use of Sentinel 1A SAR imagery to extract first a wave number direction spectrum within a region of thin sea ice in the Beaufort Sea. Physical laws are introduced that use the SAR data to get the apparent wave attenuation and wave attenuation due to ice effect. A calibration scheme is then introduced that takes the modeled wave frequency spectrum, generates the wave attenuation due to sea ice, and then compares these values to the data. The calibration scheme is used to find optimal model parameters of the shear modulus and molecular viscosity to best reproduce the observed conditions. Optimal values are presented along with a detailed discussion of how they compare to previous studies. The detailed analysis of wave attenuation required to calibrate the wave model has then also been used to discuss the physics in play. A particular emphasis here, and through the analysis is on differing characteristics within the ice pack before or after lead formation. The authors do well to show the role of their work within the science of the highly complex system of wave ice interaction and the benefits of using the analysis and calibration presented. After the authors address points i have below regarding the documentation of the papers methodology I suggest it is fit for publication.

We thank the reviewer for the support which is very encouraging.

Whilst this paper well documents the technical parts of data analysis and model calibration, and thoroughly discusses the success, limitations and implications of their results, it is severely lacking in any general description of the methods used. I had to read the entire paper, and all the specific details of the methods before I could understand the general incentives and methods of the study. I have suggestions below for how to improve the paper to address this. Once I had worked out what methods of the paper were the various sections of the paper fell into place and the science behind the paper was well founded and thoroughly discussed. However, I still have some key questions.

1) The paper does not clearly explain the reader the format of the data used and the extent of analysis performed for this study. Do you take the wave energy spectrum from Stope 2018b or the wave number direction spectrum? Or you start with the raw SAR imagery and repeat the same analysis as before? I spent much of section 2 asking myself this. Also the format of the introduction adds to this confusion, whilst it starts with 4 paragraphs of overview, paragraph 5 is neither an introduction to the area of interest, or a description of the data used in this paper. The rest of the paper has a similar ambiguity between the data sources for this study, the data processed as part of this paper and previous studies that are being cited for comparison.

2) There is little description within the paper of what is being documented by each section. The brief outline on lines 103-109 is again ambiguous to what is being performed. For example 'site description and wave characteristics are depicted in section 2' does not inform me as to whether section 2 contains description of the data used in the study and how you processed it, a summary of previous studies of the region and what they found, or even a description of a model run of the region. Section 2 itself starts immediately with a definition of kr, which is clearly done, but there is no description at all of whether kr in this study is an original data source collected by the study or a previously collected data that is being analyzed here. This lack of overview is present in near all of the technical sections with the paper. I suggest adding a clear paper overview of the form; data was obtained from (here, here, etc) that we analyze thusly to extract this information. We then ran these models and using this scheme and the previously documented data we calibrated these parameters. This information also needs to be clearly defined at the beginning of each section. Once I had worked out this information for myself I was easily able to understand the technical parts of the study, which are well written.

3) There are two key values used in this study, apparent wave attenuation and the wave attenuation due to ice effect. The notation for these needs to be changed. kr is a wave number spectrum, alpha is the associated wave attenuation, but ki is just a wave attenuation whilst previous notation leads me to think it will also be a wave number. While the methods used to get these values are documented, there is no physical description of what these values represent and why you are interested in them. A paragraph in the introduction introducing these physical values, and the other values considered in this paper (dominant wave number, wave energy spectrum for example) will greatly help a non-specialist reader. Section 3 needs an introduction explaining the differences between kr and ki, and as mentioned before, the data you use to obtain these values. Also the wave attenuation due to ice effect ki and ki^m need to have clearer definitions. I see that you have used ki as the modeled attenuation, and ki^m as that data derived value. This notation needs adjusting, the use of superscript m to define the value that is not modeled is confusing. Section 3.2 will benefit from a definition of ki at the beginning, as currently the reader has to wait till the end of the section to understand what the aim of the presented method is.

4) I suggest that the introduction section needs restructuring and expanding upon. Currently there is one paragraph on the place of wave sea-ice interaction within climate science and then a very technical description of one wave model. The paper would benefit from having first the next paragraph that explains the uses of wave forecast models, and then the technical description of WaveWatch follow. The final paragraph of the introduction needs to moved to a dedicated data description and expanded upon. It is currently unclear what SAR data used in this study is new and what has been previously published. If the SAR observational data and analysis or processing has been previously published, then a clear section citing this publication and describing the data is needed. If there is novel data work in this study then this needs to be clearly stated and the technique and data clearly described.

We apologize for the lack of sufficient details. Several major changes will be made in the final revised manuscript to address the four concerns raised above:
- Modify and expand section 2 to include all data and site description.
- Add explanation particularly to the dataset used and the analyses done in the present study.

- Clarify notations, especially the wavenumber and attenuation rates which are now defined in the Introduction.
- Revise Introduction, and the beginning of each section and subsection to clearly describe the purpose of each section.

The specifics are given below.

Specific edits:

'wave propagation' this is the first sentence in the paper. Please be more explicit in which waves you re discussing. Consider an additional sentence introducing ocean surface gravity waves 14 'fall 2015' please give a more accurate time. The study is for a single day i believe.
We will specify "ocean waves" in the title and throughout the manuscript whenever necessary. We also added the date to the SAR dataset.

- 24 this sentence is very long. I had to re-read several times to understand what became lower where. Consider splitting.
The long sentence here and several other long sentences throughout the manuscript will be split into short and clear sentences.

These citations are 10+ years old now, the reduction and predictions have changed a lot in these 10 years.
Some recently published references will be added in the introduction.

Wavewatch is first introduced with a very technical description. Can you first give an overview of Wavewatch? Its uses and the incentives behind its creation. A brief description of its main physical constructs will also be useful to the reader. The ice effects as modeled by WaveWatch are described here, but I'd like to see how these ice effects sit within the whole model.
An introduction into how wave forecast models work would be useful here. I'd like to know how to use a wave forecast model in general, then I can better understand the context of your work in the wave sea-ice interaction aspect.
A brief description of WAVEWATCH III will be added to the second paragraph in the introduction section.

Here and above you mention IC3 (and IC5). You mention that they both store and dissipate mechanical energy but what are the differences between them.
The difference between IC3 and IC5 will be explained in the Introduction. They both describe the ice cover as a linear viscoelastic material, but IC5 also uses a thin-plate assumption while IC3 does not. We will emphasise this point in the revision.

This next paragraph needs moving to a dedicated data section. Within this section it needs to be made clearer what data was used as part of this study and why. Also I find it difficult to tell from this paragraph which data was collected together in a previous study and which data was brought together for this study. Those which were brought together here needed to be expanded upon. The benefits and limitations of each data need to be better discussed.
Currently the novel method of Ardhuin 2017 and Stopa 2018a is well described, as are the reasons for its use in this paper. However there is less description for the Buoy and ice concentration data. Due to the data all being plotted together in figure 1, I am assuming that this combination of data sources is novel for this study. If this is the case please state so, and explain why the data have been chosen. If however these data have been combined before (which then explains why they are described in the introduction) then citations/descriptions of findings are needed.

We will re-organize section 2 to include all data and site description, including the paragraph in the introduction regarding Figure 1. The following points will be added.
  a) The dataset from Stopa et al. (2018b) was used as the starting point of this study. We will specify the additional analyses performed in the present study in the Abstract, Introduction, and Data description section.
  b) The data from buoys and the ship shown in figure 1(a) are not used in this study. These data sources are in the marginal ice zone with mostly pancake ice studied in Cheng et al. (2017). Showing these locations in figure 1(a) contributes to a big picture of the SAR imagery.
  c) The AMSR2 concentration and SMOS thickness data have been used in several publications in a special issue of Journal of Geophysical Research: Oceans regarding the "Arctic Sea State and Boundary Layer Physics Program". An overview of the program is in the reference list of this manuscript: Thomson et al. Overview of the arctic sea state and boundary layer physics program, Journal of Geophysical Research: Oceans, 123, 8674-8687, 2018. This reference is included in the manuscript.

Figure 1 There is no (a) label in the figure. Can you show on pane (a) how (b and c) colocate? Symbol (a) is added in figure 1, the vertical range in panel (a) is from 0 to 750 km. While in panels (b)(c), the vertical range is from 450 to 750 km. All three plots are in the range-azimuth coordinate. Hence it is straightforward to collocate them.

What is the methodology of calibrating IC3? Is it used extensively in this paper? If it is then a summary of the equational from is needed. If this is complicated then consider an appendix for the equations. If it is still too complicated then a paragraph explaining how it works with relevant citations is needed.
Section 4 describes the methodology of calibrating IC3 and IC5. To obtain the "best fit" parameters, it uses an optimization method to minimize the difference between measured and model data. Details are given in the manuscript.

Next: a full description of wave energy spectrum E(f,theta) is needed. Ether the equational form of E() needs to be added, or how it is produced by the model and how it is extracted from the data set. The source of the various E() used in this paper is a current theoretical hole for me when reading it. As it is a key value for methods of this paper, consider a paragraph in the introduction explaining how the energy spectra is calculated from observations and treated within the model you are calibrating. Some background science on ocean surface gravity waves are described, including what the wave energy spectrum is would be a useful inclusion for this paper.
Sorry for this confusion. Since E(f,theta) is not used in the study, thus will be removed in the revision. In this study, we only use wavenumber-direction spectrum from Stopa et al. (2018b) as clarified at the beginning in section 2. This study doesn't involve the process of retrieving the wave energy spectra from the observations.

Also: At this point in the paper I don't want to have to dig into section 4 to discover what modification you have had to make to the methodology of Cheng 2017. At this point I flipped to section 4 and searched for both the 'Cheng' citation and the string 'IC3' within the paper and it was still unclear to me how the methodology of Cheng 2017 worked, why you had to modify it and what those modifications are. A description of all these aspects of this study is needed, either here, or at the beginning of section 4

I started reading this paragraph hoping to understand the incentives and aims of this study and what methods were used. The paper is unlikely to have been written in a linear fashion, but please remember that the reader will read it in a linear fashion, and is likely to be unaware of all the previous work on this topic, the data you are using, the model you are calibrating and the techniques you are using to calibrate it.

We fully understand the problem the reviewer mentioned. To read a paper we also often need to go back and forth, instead of linearly. To help the readability, we will revise the Introduction to better describe the structure of this manuscript. Overviews of each section are also added at the beginning of sections 2, 3, 4 and 5.

The reference to Cheng et al. (2017) is necessary in the Introduction because both the present and the 2017 study share the same goal: to calibrate wave-in-ice models so that they can be used in global models such as WAVEWATCH III. It has been a big challenge to conduct such calibration due to lack of data. We used data in mostly pancake ice fields in the 2017 study. We are very fortunate that shortly after that we have the SAR data in the pack ice zone, so that we could expand the calibration to a different type of ice cover.

The calibration is performed by minimizing the theoretical attenuation over a spectrum of wave components with the measured data. The best-fit parameters are the calibrated model parameters. The wave-in-ice models, including IC3 and IC5, are all in terms of attenuation-frequency. Hence to do this optimization the most straightforward way is to use the attenuation-frequency data. In the 2017 paper we used buoy data. They were based on time series, hence automatically in attenuation-frequency form. In the present study the spectral attenuation derived from SAR data is in terms of wavenumber. Hence additional steps are required to map the attenuation-frequency results from the theoretical models to the attenuation-wavenumber space. This is non-trivial, but an intricate point that needs to be pointed out. We will explain the methodology better in the revision. The Introduction only briefly mentions that the methodology is not as straightforward as in the 2017 study.

I am assuming that this section is a description of the data used in the study. However you give no indication of this in the text. I start by asking 'the wave number spectrum retrieved from where?'. This does not get answered, so the following information is very difficult to understand.

This range suggests you are talking about figure 1? I should have to work this out myself.

fitted to which E(f,theta) from where?

I can't understand the direction sum of E(kr,theta) when you have not described where kr comes from and what E(f,theta) is and where that comes from. Was there a previous analysis performed to get E(f,theta)? Was that done as part of this study or previously?

Again I am asking whether this work of extracting kr,dominant is from this study. Was it previously done as part of Stopa 2018a/b ? Please inform the reader what work was done with the data for this study.

We will revise section 2 to address the issues above.

Difficult to see the variation with concentration in the figure.

The description of the change of dominant wavenumber will be revised.

Again difficult to see the decrease with kr on the figure. Also it is unclear what you are referring to with 'wave propagation direction'.

The dominant wavenumber $k_{r,dominant}$ is indicated by the cell colors. Since the wave energy $E(k_r,\theta)$ is a two-dimensional quantity, each of these wavenumbers including the dominant one has a directional energy spectrum $E_{kr}(\theta)$. The main direction with contains most of the energy as described in the first paragraph of section 2 is shown by the arrows in the figure 1(a). Hopefully the revision clearly explains these points.

what is the meteorological convention?
The definition will be added.

126. Please give a value here for lambda_c and/or the associated limit for kr.
We will add the following comments to address this issue.
"Because the azimuth cutoff $\lambda_c$ is estimated from the SAR image, $\lambda_c$ changes at different locations. Stopa et al. (2018b) gave two examples of $\lambda_c$ at different locations. That is, $\lambda_c \approx 50$ m at the drifting buoys (blue dots in figure1(a)). While at the AWAC located further into the sea ice (black asterisk in figure1(a)), $\lambda_c = 125$ m estimated from SAR image but $\lambda_c = 60$ m estimated from AWAC measurement. Thus, $k_r$ is limited by $k_r \leq \min(2\pi/\lambda_c, 0.045\text{m}^{-1})$."

Is the frequency f estimated for all kr?
f and kr are related through the open water dispersion relation as mentioned in section 3.2.

Section 3
Again this section really needs an introduction. I had to read the whole paper before I could make sense of it. Even so I still have some major questions that are easily answered by reading the paper.
We will revise this section by adding a brief introduction in the beginning of the section and subsections.

please define the 'Pearson correlation coefficient'.
Definition will be added after 'Pearson correlation coefficient'.

Power spectral density of what? I see this is mentioned in the figure, but please also define it in the text.
Mathematical definition will be added after the power spectra density.

Figure 4. It would be nice to have an overlay square in figure 1 showing how the left column of figure 4 aligns.

Because the left panel in figure 4 is in longitude-latitude coordinate system, but figure 1 is in the range-azimuth coordinate system, it makes sense to add boxes in figure 1(a), but it would make the figure too messy. Thus, instead of drawing an overlay square in figure 1(a), we redraw the latitude lines corresponding the longitudes mentioned in figure 4 to help readers identify the three subregions of figure 4 in figure 1.

Please also note in the caption that the centre column is generated by pairs of data, whilst the rightmost comes from individual points.
In the right column, data of power spectra density (PSD) curves are averaged values rather than from individual points direction. Specifically, we collect the PSDs from the end locations of pairs shown in the left column. These data are averaged values per 0.1 degree in latitude. This will be clarified in the revision.

224-229 I suggest that a version of this paragraph come at the beginning of the section. This would much ease the reading of this paper. Here I can understand clearly which values you are obtaining from where, why you chose them and what you intend to do with them. However the picture is still not entirely clear. This style of explanation is needed for every section and sub section.
The suggested brief introductions will be added to the beginning of sections 2 to 5.

and figures 3 and 5. A better description of these two figure and what you wish to show with them is needed. The caption for figure 5 is not sufficient. Saying that alpha vs kr is similar to ki^m vs kr is misleading, as alpha and ki^m are not similar physical quantities.
More descriptions will be added to explain the difference between alpha and ki. The caption for figure 5 will be rewritten.

Here you say that kr and ki re solved for, but elsewhere in the paper kr is given as a data source whereas ki is a model variable. Is kr now referring to something else?
Definition of kr and ki will be added in the Introduction.

k = kr + i ki, how are you combining a wave number from data and an attenuation factor?
Idem

You have now introduced kr^m. What is this symbol for. The m would indicate 'model' but the opposite is true for ki. However previously kr was from a data source. Please clear up all the ki kr notation. Avoid using ^m if you're not going to use it to indicate 'model' Equation 6 Is the subscript 2 supposed to be an exponent? If not what function does it refer to?
We will clean up the notations in the manuscript

I find the discussion of ki here confusing due to the mixed notation used. Why not ki^m? Also here you talk about ice effect wave attenuation, but earlier you talk about kr,dominant which is a wave number. Please can you clear up the notation for these.
Idem is the alpha here the same as in equation 1? Are you using alpha from the SAR data source in these equations to calibrate the model? If yes, then clearly say so, if not, then an alternative notation is required.
Sorry for the confusion. $\alpha$ here is not the same as in Eq. (1). Letter 'a' will be used in Eq. (5) and its description to avoid confusion.

This sentence is an important one for this paper and needs to be clearly stated in the introduction.
This information will be added in the last paragraph of the Introduction.

This information would be better suited at the beginning of section 4, before equations 5.
We will revise the text to address the issue.

are you actually running WaveWatch as part of the calibration scheme? If so which out put variable are you taking for the calibration? I am assuming that you take the model out put frequency spectrum f, and use the relevant equations from 5 to create the required ki kr.

WaveWatch III (WW3) is not used in this analysis. But we use the governing equation (which is a general one in all global wave forecast models) in WW3 to extract the wave attention coefficient due to the ice effect. To do so, we need to calculate all other source terms $S_{in}$, $S_{ds}$, $S_{nl}$. They are based on the formulations given in the WW3 manual. In addition, we also need to argue that time derivation in this governing equation is negligible as discussed in the manuscript.

I would like to see a figure of the clustering of retrieved parameters. The table plus description does not give enough information.

Moving figure S2 and S3 to the main context will make a very long manuscript. We think the scatter plots are lower level information than the other figures. Hence we keep them in the supplemental material.

Ah I see the scatter plots are in the supplementary material.       Idem

Again put the scatter plots here, they are interesting. Figure 6.       Idem

They would be a worthwhile inclusion in the main paper Table 1. Please expand the caption to say what these numbers are. They are the results of the model calibration I assume?

The caption of table 1 will be revised accordingly.

311-312 Sentence unclear please revise

We will revise the text to address this issue.

Pane b illegible and either needs redrawing in color, omitting 90% of the lines, or even not including in the paper.

Because hundreds of lines are superimposed in panel b, it looks like a black band. The band shows the wavenumber range predicted by the WS model, implying the validity of the model predicted wave dispersion using the calibrated parameters. We could use a broad band to show the same information, but it would lose some details such as the fluctuations of the curves, if the fluctuations are outliers or general, and the number of cases studied. We note that their fluctuations are within a very small range less than 5%.

Caption you mention 'the data' plotted in plot a, which data? from where? section 3 I guess.

The caption in figure 6 will be revised to clarify the data sources.

It is important to also mention here the limitations of using a numerical (or a more specific description of the algorithm used) parameter search/optimisation scheme. For example: no. minima for equation 6 could be found for the data at these locations.

The issue of implementing the numerical algorithm will be added. This genetic algorithm is commonly used for global optimization.

You here introduce a data mask, though this was not previously mentioned. Does it relate to the previous sentence on data quality? In which step of the data processing was this mask used?

Since the data 'mask' is not used and caused confusion in this study. This sentence will be deleted.

Which function are you talking about? Equation reference?

It is referred to the objective function in the model parameter optimization, i.e., Eq. (6), which will be clarified in the revision.

Can you point me back to the results to show the difference in dispersion.

It is shown in the right panel in figure 6 and figure S4, which will be added in the revision.

Appendix I suggest moving the appendix to the main paper section. The results here appear to be as worthwhile as others discussed. Are these results from another study and you include them in an appendix for reference?

We note that in the revision Appendix A will be added to explain the first appearance of leads. Hence Appendix I will become Appendix B. Appendix B shows another method in determining the wavenumber-dependent attenuation, to compare with the analysis of the attenuation of the significant wave height shown in Stopa et al. (2018b).

The method in Appendix B is different from the one in section 3. The obtained attenuation rate represents an average attenuation over the entire latitude domain covering different ice conditions, thus, cannot be used in calibration for different ice types in section 4. Therefore, bringing the appendix to the main body could be distracting to the focus of this study.